# Learning Mixtures of Continuous-Time Markov Chains

## ABSTRACT

Sequential data naturally arises from user engagement on digital platforms like social media, music streaming services, and web navigation, encapsulating evolving user preferences and behaviors through continuous information streams. A notable unresolved query in stochastic processes is learning mixtures of continuous-time Markov chains (CTMCs). While there is progress in learning mixtures of discrete-time Markov chains with recovery guarantees [24, 28, 40], the continuous scenario uncovers unique unexplored challenges. The intrigue in CTMC mixtures stems from their potential to model intricate continuous-time stochastic processes prevalent in various fields including social media, finance, and biology.

In this study, we introduce a novel framework for exploring CTMCs, emphasizing the influence of observed trails' length and mixture parameters on problem regimes, which demands specific algorithms. Through thorough experimentation, we examine the impact of discretizing continuous-time trails on the learnability of the continuous-time mixture, given that these processes are often observed via discrete, resource-demanding observations. Our comparative analysis with leading methods explores sample complexity and the trade-off between the number of trails and their lengths, offering crucial insights for method selection in different problem instances. We apply our algorithms on an extensive collection of Lastfm's user-generated trails spanning three years, demonstrating the capability of our algorithms to differentiate diverse user preferences. We pioneer the use of CTMC mixtures on a basketball passing dataset to unveil intricate offensive tactics of NBA teams. This underscores the pragmatic utility and versatility of our proposed framework. All results presented in this study are replicable, and we provide the implementations to facilitate reprodubility.

## CCS CONCEPTS

• **Mathematics of Computing**;

## KEYWORDS

Markov chains, machine learning, graph algorithms, experimentation

**ACM Reference Format:**

Anonymous Author(s). 2018. Learning Mixtures of Continuous-Time Markov Chains. In *Proceedings of Make sure to enter the correct conference title from your rights confirmation emai (Conference acronym 'XX)*. ACM, New York, NY, USA, 15 pages. https://doi.org/XXXXXXX.XXXXXXX

## 1 INTRODUCTION

Continuous-Time Markov Chains (CTMCs) are a fundamental class of stochastic processes with a wide array of applications across various domains. They are particularly crucial in modeling systems where events occur continuously over time, such as in queueing theory [39], finance [27], understanding disease progression [6, 31, 37] and telecommunications among others [41]. The inherent memoryless property of CTMCs, where the future behavior of the system is independent of the past given the present, makes them a natural choice for modeling random processes evolving over time [36]. In the realm of biological systems, for instance, CTMCs have been instrumental in modeling the stochastic behavior of genetic networks and the evolution of species [49]. Similarly, in finance, they are employed to model various continuous-time financial models including models for option pricing [27]. CTMCs for molecular kinetics have gained popularity in recent years, as they approximate the long-term statistical dynamics of molecules using a Markov chain over a discretized partition of the configuration space [38]. It is notable that in such applications, unlike our focus where we have a set of $n$ well-defined states, elucidating the state-space description isn't straightforward. The flexibility and analytical tractability of CTMCs, along with their capability to provide insightful analytical results, make them an indispensable item in the toolkit of researchers and practitioners dealing with stochastic systems evolving over time. In this study, we delve into the largely untouched realm of learning mixtures of homogeneous CTMCs from trail data (refer to Section 2 for a formal definition).

Formally, a mixture $\mathbf{M}$ is represented by a tuple of $L \geq 2$ Markov chains on a finite state space $[n] := \{1, \ldots, n\}$, denoted as a sequence $\mathbf{M} = (M^1, M^2, \ldots, M^L)$. Each chain is linked with a vector of initial probabilities, denoted as $s^\ell \in \mathbb{R}^n$ with $\sum_{\ell=1}^{L} \sum_{y=1}^{n} s_y^\ell = 1$. For discrete-time Markov chains, each chain $\ell \in [L]$ is defined by a stochastic matrix representing transition probabilities. The aim is to ascertain the parameters of $\mathbf{M}$, encompassing the transition matrices and initial probabilities, based on observed trail data [24, 40]. In the case of continuous-time Markov chains (CTMCs), each chain is now characterized by a *rate* matrix $K^\ell$ along the starting probabilities. The continuous-time stochastic process unfolds as follows: Initially, we find ourselves in a chain $\ell \in [L]$ and state $y \in [n]$ with a probability of $s_y^\ell$. Subsequently, the transition between states is directed by the rate matrix $K^\ell$. Specifically, in state $y$, we select exponential-time random variables $E_z \sim \text{Exp}(K_{yz})$ for all states $y \neq z$. We then transition to state $z^* = \arg\min_z E_z$ after a time duration of $E_{z^*}$. Upon transitioning states, we reiterate the process. This mechanism generates a trail $(x_t)_{t \geq 0}$ where $x_t$ represents the state at time $t$.

The challenge entails the recovery of a mixture of CTMCs, specified as follows: Provided a set $X$ of continuous-time trails $(x_t)_{t \geq 0}$, is it possible to retrieve the rate matrices $\mathbf{K} = (K^1, \ldots, K^L)$ and starting probabilities $(s^1, \ldots, s^L)$, albeit up to a permutation?

In practical scenarios, the observation of a continuous-time process is typically approximated through discrete-time observations.

A significant challenge, which inherently does not arise in the recovery of discrete-time Markov chains, entails the recovery of continuous-time chains from their discretized observations. Numerous previous works have addressed this issue, such as [33] through Maximum Likelihood Estimation (MLE) or by computing the $p$-th root of the matrix exponential [25]. In this study, we extend the efficient MLE approach of [33]. Intuitively, a primary difficulty lies in the fact that due to discretization, some states remain unobserved. Additional known challenges include the admissibility of the rate matrix under noise [33]. These challenges become more pronounced when recovering a mixture of CTMCs, as additional noise, for instance from incorrectly assigning a trail to the incorrect chain, comes into play. A direct formulation of the mixture problem as MLE is inefficient as solving the MLE for a mixture involves the posterior probabilities of each observed trail, leading to numerous inter-dependent terms (cf. Section 3). Remarkably, we demonstrate how to employ the MLE of [33] to recover the mixture while maintaining its efficiency through soft clusterings.

Fortunately, recent works have addressed the task of learning mixtures of discrete-time Markov chains, providing a foundation to learn the discrete-time mixture for any specified discretization interval under certain lenient conditions [24, 40]. However, the hurdle of recovering mixtures of Continuous-Time Markov Chains (CTMCs) still stands: transitioning from discrete to continuous-time chains introduces additional challenges compared to the discrete-time scenario, primarily due to the variance in transition rates, as discussed in detail in Secton 3.1.

In this work, we make the following key contributions:

• We present a versatile algorithmic framework for learning mixtures of continuous-time Markov chains through both continuous and discrete-time observations and methods to tailor it depending on the length of the trails.

• We illustrate how the observed trails' length and the mixture parameters lead to varying problem regimes. Each regime demands distinct algorithms, for which we provide recommendations. Our recommendations rely on our theoretical results that represent new contributions and use advanced probabilistic tools, including Chernoff-like bounds for Markov chains [15].

• We explore the effects of discretizing continuous-time trails on the learnability of the continuous-time mixture. As continuous-time processes are usually observed through discrete, costly observations, selecting the right discretization is crucial.

• We conduct a thorough experimental analysis, contrasting our methods with leading-edge competitors. Experimentally, we delve into the examination of sample complexity, exploring the trade-off between the number of trails and their lengths. Our discoveries offer valuable insights that can assist in selecting a suitable method for a given problem instance. We apply our algorithmic framework on user-generated trails from the Last.fm platform, demonstrating its ability to effectively capture users' listening patterns.

• We introduce *Markovletics*, a novel application of mixtures of Markov chains. Specifically, we pioneer the use of CTMC mixtures on a basketball passing dataset that unveils offensive strategies of NBA teams.

• In Appendix 5, we present thorough proofs and a broader set of experiments, shedding light on the advantages and limitations

**Table 1: Frequently used notation.**

| Symbol | Definition |
|---|---|
| $n, L \in \mathbb{N}$ | number of states and chains resp. |
| $(x_t)_{t \geq 0}$ | continuous-time trail with $x_t \in [n]$ |
| $\tau > 0$ | discretization time parameter |
| $\mathbf{x} \in [n]^m$ | discrete-time trail of length $m$ |
| $X (X^\ell)$ | set of $r$ continuous-time trails (from chain $\ell$) |
| $\mathbf{X} (\mathbf{X}^\ell)$ | set of $r$ discretized trails (from chain $\ell$) |
| $c_y^\ell$ | # of transitions over $y$ from trails in $\mathbf{X}^\ell$ |
| $K_{\min}$ and $K_{\max}$ | min and max rate in the mixture $\mathbf{K}$ |

of both the proposed methods and the baselines. Additionally, for the sake of reproducibility, our code can be accessed publicly at [1].

## 1.1 Definitions and Notation

For ease of reference, we summarize the notation in Table 1. We already introduced a CTMC as a continuous stochastic process over the states $[n]$ where each transition is governed by a rate matrix $K \in \mathbb{R}^{n \times n}$. For the reader's convenience, a refresher on the definition of a CTMC can be found in Appendix A. We can define the rate matrix through the limit of its discretization. Let $Y(t) \in [n]$ for $t \geq 0$ be a random variable that holds the state of the CTMC at time $t$. We define $T_{yz}(\tau) = \Pr[Y(t + \tau) = z \mid Y(t) = y]$ for any $t \geq 0$. In this work, we always assume that the CTMC is time-homogeneous, which makes $T_{yz}(\tau)$ independent of $t$, due to the memorylessness of the process. Note also that $T(\tau)$ is the transition matrix of the discrete-time Markov chain arising from observing the CTMC at regular time intervals $\tau$. We obtain the rate matrix as $K = \lim_{\tau \to 0} \frac{1}{\tau}(T(\tau) - I_n)$ where $I_n \in \mathbb{R}^{n \times n}$ is the identity matrix. By this definition, the diagonal elements are $K_{yy} = -\sum_{z \neq y} K_{yz}$ and describe the distribution of the time $t$ required to transition to any state. We call the mean of $t$ the holding time of state $y$. Conversely, we obtain the discretization through the matrix exponential $T(\tau) = e^{K\tau}$ [30, 33, 36]. As discussed, a trail is a sequence $(x_t)_{t \geq 0} = Y(t)$ for a fixed sample of the random variable $Y$. For a mixture $\mathbf{K}$, we define as $X^\ell$ as a set of sampled trails from $K^\ell$ and the set of all trails as $X = X^1 \cup \cdots \cup X^L$.

To define the distance between two CTMCs, we look at the distribution generating the next state. That is, for a fixed state $y \in [n]$, we consider the distribution over $(z, t)$ where $z$ is the next state and $t$ the time of this transition. Here, $z$ and $t$ are as defined in the previous section through the minimum of a set of exponential random variables. We can evaluate the distance of two CTMCs $K$ and $K'$ in state $y$ through the total variation distance (TV-distance) of the distribution over the tuples $(z, t)$ [30]. The TV-distance evaluates to

$$\text{TV}(K_y, K'_y) := \frac{1}{2} \sum_{z \neq y} \int_0^\infty |K_{yz} e^{t K_{yy}} - K'_{yz} e^{t K'_{yy}}| dt.$$

We further define the *recovery error* between two CTMCs as the average of the above TV distance from all states

$$\text{recovery-error}(K, K') := \frac{1}{n} \sum_x \text{TV}(K_x, K'_x).$$

We then define the recovery error between two mixtures $\mathbf{K}$ and $\mathbf{K}'$ as the cost (wrt. the recovery-error on CTMCs) of a minimum

assignment between the chains in the mixture

$$\text{recovery-error}(\mathbf{K}, \mathbf{K}') \coloneqq \frac{1}{L} \min_{\sigma \in S_L} \sum_{\ell=1}^{L} \text{recovery-error}(K^\ell, (K')^{\sigma(\ell)})$$

where $S_L$ is the symmetric group of all permutations on $[L]$.

We always assume that the CTMC $K$ is irreducible, such that it has a (unique) stationary distribution $\pi_K$. The mixing time $t_{\text{mix}}(K)$ of a CTMC $K$ is defined as the smallest time $t \geq 0$ such that $\text{TV}(sT(t), \pi_K) \leq 1/3$. Analogously, the mixing time $t_{\text{mix}}(M)$ of an ergodic, discrete-time Markov chain $M$ is the smallest integer $t \geq 0$ such that $\text{TV}(sM^t, \pi_M) \leq 1/3$. Note that $\lceil t_{\text{mix}}(K)/\tau \rceil = t_{\text{mix}}(T(\tau))$.

## 2 RELATED WORK

The exploration of Markov chains constitutes a core subject within the realm of probability [5, 19, 30, 36] and computer science [10, 13, 14, 17, 20, 22, 26, 34]. We focus on work that lies closest to ours.

**Learning Mixtures of Discrete Markov Chains.** The problem of learning mixtures of discrete Markov chains has been well studied in the literature. The Expectation Maximization (EM) algorithm [18, 48] can be employed to locally optimize the likelihood of the mixture, or one could learn a mixture of Dirichlet distributions [43] albeit without solid theoretical assurances regarding the output quality. On the other hand, moment-based techniques, leveraging tensor and matrix decompositions, can be used to provably learn (under specified conditions) a mixture of Hidden Markov Models [3, 4, 42, 43] or Markov Chains [43, Section 4.3]. The approach introduced by Gupta et al. [24] stands out as markedly more efficient and scalable compared to the latter techniques, which are dependent on 5-trails, thereby causing their sample complexity to increase as $n^5$ instead of $n^3$. Spaeh and Tsourakakis [40] delved deeper into these conditions, illustrating that they dictate constraints on the connectivity of the chains. However, they demonstrated that these constraints can be relaxed for a wider class of chains, which remain learnable despite the eased conditions.

**Learning Mixtures of homogeneous CTMCs.** Extensive research has been conducted on learning a single continuous-time Markov chain from trails, yet transferring these techniques to handle mixtures presents a challenging and relatively uncharted domain. Bladt and Sørensen [9] introduced an EM algorithm, along with methodologies for certain special cases, including scenarios where the rate matrix $K$ is diagonalizable. However, this condition is often too restrictive for practical real-world applications [31]. Various methods have been explored and experimentally evaluated as documented by Tataru and Hobolth [46], revealing that these methods essentially compute differently weighted linear combinations of the expected values of the sufficient statistics. On a related note, McGibbon and Pande [33] devised an efficient Maximum Likelihood Estimation (MLE) technique to recover a single CTMC from sampled data. Two distinct variants have been proposed: one for learning a general CTMC and another for a reversible CTMC that adheres to the stated balance equations [33, 38]. The latter is tackled as a constrained optimization problem. As previously highlighted, there remains a conspicuous absence of algorithms with recovery guarantees, specifically crafted for mixtures of CTMCs that broaden the framework of Gupta et al. [24] into the continuous domain. This void is presumably a result of the inherent intricacy

of the challenge, amplified by the nuanced hurdles presented by continuous-time processes compared to their discrete-time analogs. A line of inquiry that aligns closely with the current discourse is that of Continuous-Time Hidden Markov Models (CT-HMM) [11]. Within a CT-HMM framework, both the hidden states (akin to a traditional HMM) and the transition times marking the alterations in hidden states remain unobserved. CT-HMMs manifest as a particular instance of continuous-time dynamic Bayesian networks [35], wherein the EM algorithm [8] is employed. Luo et al. [32] have advanced a Markov Chain Monte Carlo (MCMC) methodology for deducing a mixture of CT-HMMs [32]. However, as elucidated by Liu et al. [31], this avenue of investigation grapples with scalability constraints. In response, more scalable strategies rooted in CTMCs have been formulated by Liu et al. [31].

**Time parameter $\tau$.** The parameter $\tau$, also referred to as time lag or discretization parameter in other contexts, is crucial in the discretization of CTMCs. Intuitively, a "too small" value of $\tau$ results in a scenario where no transitions are observed, while a "too large" $\tau$ may lead to the observation of numerous transitions, many of which are not direct. In other words, by ranging $\tau$ from 0 to $+\infty$ we obtain a sequence of count matrices $C(\tau) \in \mathbb{R}^{n \times n}$ where $c_{yz}(\tau)$ is the number of transitions from $y$ to $z$ within time $\tau$. Determining the appropriate scale for a single CTMC is a complex task for which sophisticated techniques have been devised. One common approach in molecular kinetics is the use of implied timescales, a method initially introduced by Swope, Pitera, and Suits [45]. Nonetheless, this method operates as a heuristic and is burdened by significant computational expenses due to the necessity of computing eigenvalues for a sequence of transition matrices $T(\tau_k = k\Delta t)$, for several integer values of the variable $k$.

**Choosing the number of chains $L$.** One strategy involves utilizing model selection indices such as the Akaike Information Criterion (AIC) or the Bayesian Information Criterion (BIC) [21]. Another well-known strategy is the elbow method [21, 29, 44]. A more theoretically grounded technique was introduced recently by Spaeh and Tsourakakis [40], who leverage restrictions on the singular values of certain matrices to inform the selection of $L$. In this work, we assume $L$ is part of the input for the theory part, but we experimentally evaluate this choice.

## 3 PROPOSED METHODS

**Algorithmic framework.** In this section, we present our framework for learning mixtures of CTMCs, using continuous-time or discretized trails (Algorithm 1). Our framework comprises three stages: discretization, soft clustering, and recovery. The advantage of this division is that each phase can be tailored independently based on the characteristics of the mixture under study and the sampling process. As a key characteristic of the latter, we use the length of the trails $m$. A comprehensive description of these three stages is provided in the following sections. On a high level, we first discretize the continuous-time trails by observing each trail at regular time intervals. Note that in practice, this step may be part of the data-generating process whenever continuous observation is impossible or too costly. Below, we provide rules on setting the discretization parameters based on the properties of the mixture. In the second step, we learn a *soft* clustering based on the discretized

trails which assigns each trail to a chain in a probabilistic manner. Here, we employ techniques developed for learning mixtures of discrete-time Markov chains. Finally, we use a maximum-likelihood estimate base on the soft clustering to recover all chains.

---

**Input:** Set of $r$ continuous-time trails $X$, discretization rate $\tau$, number of chains $L$
**Output:** Continuous-time mixture $\mathbf{K} = (K^1, \ldots, K^L)$ and starting probabilities $(s^1, \ldots, s^L)$
Let $\mathbf{X} = \{\mathbf{x} = (x_{i\tau})_{0 \le i \le r} \in [n]^m : x \in X\}$
Learn a soft assignment $a \colon \mathbf{X} \times [L] \to [0, 1]$ from $\mathbf{X}$
**for** $\ell = 1, \ldots, L$ **do**
  Let $s_y^\ell = \frac{1}{r} \sum_{\mathbf{x} \in \mathbf{X} : \mathbf{x}_0 = y} a(\mathbf{x}, \ell)$ for all $y \in [n]$
  Learn $K^\ell$ using MLE on $\mathbf{X}$ with weights $\{a(\mathbf{x}, \ell) : \mathbf{x} \in \mathbf{X}\}$
**end**
**Algorithm 1:** Framework for Learning Mixtures of CTMCs.

---

We defer all proofs from this section to the Appendix, where they are grouped by subsection.

## 3.1 Discretization

A continuous-time trail $(x_t)_{t \ge 0}$ is discretized by observing it $m$ times at regular time intervals $\tau > 0$ [33]. That is, a sequence of states $\mathbf{x} = (x_{i\tau})_{0 \le i < m}$ is obtained with $\mathbf{x}_i = x_{i\tau} \in [n]$ for all $0 \le i < m$. Discretization is necessitated by our methods, but it is frequently needed when the stochastic process cannot be observed continuously [33]. In many real-world scenarios, observing a stochastic process at a fixed time is costly and thus subject to budget constraints. Given control over the observation times, it is thus an important task to optimize the discretization parameters $\tau$ and $m$ to enable the best possible learning of $\mathbf{K}$. To learn a mixture, two main steps are necessary: (1) clustering the trails and (2) estimating the parameters of the exponential random variables of the underlying continuous-time stochastic process. Concerning (1), it is crucial for each trail to be lengthy enough to discern the distinct model differences among the chains within the mixture. We quantify this by providing clustering guarantees for (1) in Section 3.2. For (2), the appropriate selection of $\tau$ is vital as we already explained in Section 1 and 2. We now present certain criteria, independent of the subsequent clustering method, to choose the discretization rate $\tau$. Specifically, we discuss the challenges in choosing $\tau$ by introducing a basic estimator for the rate matrix $\hat{\mathbf{K}}$ of the underlying CTMC under the assumption that we have a correct clustering. We will see how the choice of $\tau$ and estimation quality depends on the structure of the rate matrix $\mathbf{K}$. These structural challenges are not an artifact of our estimator, but pertain for other estimators, as we verify experimentally in Section 4.

For each chain $\ell \in [L]$ and state $y \in [n]$, we break down the estimation process of a single row of the rate matrix $K_y^\ell := (K_{yz}^\ell)_z$ into two phases. Initially for each state $y$ we estimate the rate $|K_{yy}^\ell|$ (it is essential to recall that by definition in Section 1, the diagonal of $K$ is negative). In order to define our estimator, let $c_y^\ell = \{\mathbf{x} \in \mathbf{X}^\ell, 0 \le i < m : \mathbf{x}_i = y\}$ be the number of times we transition through $y$ in the set $\mathbf{X}^\ell$ of trails from chain $\ell$. We estimate the rate $|K_{yy}^\ell|$ through the holding probability $q_y^\ell := e^{|K_{yy}|\tau}$ and

thus define the estimators

$$\hat{q}_y^\ell := \frac{1}{c_y^\ell} |\{\mathbf{x} \in \mathbf{X}^\ell, 0 \le i < m : \mathbf{x}_i = y \land \mathbf{x}_{i+1} = y\}| \quad \text{and}$$

$$\hat{K}_{yy} := \frac{1}{\tau} \log(\hat{q}_y^\ell).$$

With an appropriate universal choice of $\tau$ (discussed in the proof), we obtain the following estimation guarantee:

LEMMA 1. *Let $0 < \epsilon_h < 1$ and fix a state $y$ and chain $\ell \in [L]$. With $c_y^\ell = \Omega\left(\epsilon_h^{-2} \log(Ln)\right)$ transitions, our estimator $\hat{q}_y^\ell$ for the holding time satisfies $|\hat{q}_y^\ell - q_y^\ell| \le \epsilon_h q_y^\ell$ with high probability.*

Note that we consider all consecutive steps $(i, i+1)$ as a transition, even though there may not be a state change. Second, we estimate the transition probabilities $p_{yz}^\ell$ from $y$ to another state $z$ through

$$\hat{p}_{yz}^\ell := \frac{|\{\mathbf{x} \in \mathbf{X}^\ell, 0 \le i < m : \mathbf{x}_i = y \land \mathbf{x}_{i+1} = z\}|}{|\{\mathbf{x} \in \mathbf{X}^\ell, 0 \le i < m : \mathbf{x}_i = y \land \mathbf{x}_{i+1} \ne z\}|}$$

$$\hat{K}_{yz}^\ell := \hat{p}_{yz}^\ell |\hat{K}_{yy}^\ell|.$$

The quality of estimation of the transition probabilities is detailed in Lemma 2 in Appendix B. Our estimation critically optimizes the following trade-off: As we increase $\tau$, some direct transitions become unobservable. On the other hand, excessively reducing $\tau$ results in considerable redundancy and challenges with numerical stability. Informally, we desire to minimize the number skipped intermediate transitions due to the $\tau$ time resolution. We define the notion of a bad transition as follows:

DEFINITION 1 (BAD TRANSITION). *A pair $(x, i)$ of a continuous-time trail $x$ and a step $0 \le i < m$ is called a bad transition if $x_{i\tau} = y$, $x_{i\tau+\zeta} = z$ for a $\zeta \in (0, 1)$, and $x_{i\tau+\tau} = y'$, for states $y \ne z$ and $z \ne y'$.*

Our estimators' quality deteriorates as the number of bad transitions increases and we therefore aim to keep the number of bad transitions small. Clearly, the probability to obtain a bad transition is maximized for the state with maximum rate $K_{\max} := \max_{\ell, y} |K_{yy}^\ell|$, and we can show (cf. Lemma 1) that

$$\Pr[\text{bad transition}] \le \min(1, K_{\max}^2 \tau^2),$$

which motivates setting $\tau$ inversely proportional to $K_{\max}$ in order to keep the fraction of bad transitions to a small constant. Disregarding bad transitions, the estimation's quality as defined by Lemma 1 is primarily determined by the total duration during which we observe the holding time without transitions. For one trail $x \in X^\ell$, this is as follows:

$$\tau \cdot |\{0 \le i < m \mid x_{i\tau+\zeta} = y \text{ for all } \zeta \in [0, \tau]\}|$$

We aim to maximize the expectation of this term, over all transitions, which by the memorylessness of the exponential random variable $E \sim \mathrm{Exp}(|K_{yy}^\ell|)$ and the Markov process is

$$\tau \sum_{x \in X^\ell, 0 \le i < m} \Pr[x_{i\tau} = y \land E > \tau]$$

$$= \tau \Pr[E > \tau] \sum_{x \in X^\ell, 0 \le i < m} \Pr[x_{i\tau} = y] = \tau \mathbb{E}[c_y^\ell] e^{K_{yy}^\ell \tau}.$$

Similarly, to prove Lemma 2, it is essential to optimize the count of observed transitions with state changes to ensure accurate estimation of transition probabilities. In expectation, the number of transitions from state $y$ resulting in a state change is equal to

$\mathbb{E}[c_y^\ell](1 - e^{K_{yy}^\ell \tau})$ which is the lowest for $K_{\min} := \min_{\ell, y} |K_{yy}^\ell|$. We detail the behavior of both quantities and the number of bad transitions in Figure 6 (in Appendix A). It is evident that we require large values of $\tau$ to efficiently capture the exponential random variables dictating the state transitions. In particular, the discrepancy in estimating holding times and transition probabilities motivates us to define the *condition number* of a continuous-time Markov chain with rate matrix $K$ as $\kappa := \frac{K_{\max}}{K_{\min}}$ where $K_{\max} = \max_{\ell, y} |K_{yy}^\ell|$ and $K_{\min} = \min_{\ell, y} |K_{yy}^\ell|$. To estimate the CTMC $K^\ell$ for any $\ell \in [L]$, and $0 < \epsilon < 1$, we set $\tau := \frac{\epsilon}{100\kappa K_{\max}}$ and obtain the following theorem:

**Theorem 1.** *Fix a chain $\ell \in [L]$ and a state $y \in [n]$. If the number of transitions is $c_y^\ell = \Omega\left(\frac{\kappa^2}{\epsilon^3}\left(n + \frac{\kappa^2}{\epsilon}\right)\log(Ln)\right)$, we can obtain $\hat{K}_y^\ell$ such that $\mathrm{TV}(K_y^\ell, \hat{K}_y^\ell) \le \epsilon$ with high probability.*

Consistent with our earlier discussion, the number of transitions needed increases with $\kappa$. This necessitates setting $\tau$ at a sufficiently small value to steer clear of bad transitions. Consequently, a larger sample set is required to effectively gauge the rates and transitions of states possessing lower rates. Following this, we derive the subsequent corollary, under the presumption that the number of transitions for each chain and state are close to uniform and we are aware of the underlying chain for each trail:

**Corollary 1.** *If $c_x^\ell = \Omega(\frac{rm}{Ln})$, then we obtain an estimator $\hat{K}$ of $K$ with recovery-error$(\mathbf{K}, \hat{\mathbf{K}}) \le \epsilon$ using a total of $r$ trails where $r = \Omega\left(\frac{Ln}{m} \cdot \frac{\kappa^2}{\epsilon^3}\left(n + \frac{\kappa^2}{\epsilon}\right)\log(Ln)\right)$ with high probability.*

In practice, we frequently do not have knowledge of the underlying chain for each transition. The subsequent section delves into strategies to address this challenge.

## 3.2 Soft Clustering

In this section, we discuss how to assign each trail to a chain, in a soft (i.e., probabilistic) manner. Specifically, we aim to learn a soft clustering $\hat{a}: \mathbf{X} \times [L] \to [0, 1]$ such that $\hat{a}(\mathbf{x}, \ell)$ is approximately proportional to the probability of generating $\mathbf{x}$ with the $\ell$-th chain $K^\ell$. We denote this probability as $\Pr[\mathbf{x} \mid \mathbf{K} \cap \ell]$ where, in an abuse of notation, we write $\mathbf{x}$ for the event that the discretized trail $\mathbf{x}$ is generated from the mixture $\mathbf{K}$ and use $\ell$ for the event that we choose the $\ell$-th chain in $\mathbf{K}$. Formally, we set

$$a(\mathbf{x}, \ell) := \frac{\Pr[\mathbf{x} \mid \mathbf{K} \cap \ell]}{\sum_{\ell'} \Pr[\mathbf{x} \mid \mathbf{K} \cap \ell']} \qquad (1)$$

and want that $\hat{a}(\mathbf{x}, \ell) \approx a(\mathbf{x}, \sigma(\ell))$ for all $\ell \in [L]$ and $\mathbf{x} \in \mathbf{X}$ under some fixed permutation $\sigma \in S_L$. We call such a soft clustering (approximately) valid. In the following Section 3.3, we will argue formally that a valid soft clustering is important for the recovery of the CTMCs. To obtain such a valid soft clustering, we utilize techniques developed for learning mixtures of discrete-time Markov chains and use the simple fact that for the discretized mixture $\mathbf{T}(\tau) := (e^{K^1\tau}, \dots, e^{K^\ell\tau})$ holds $\Pr[\mathbf{x} \mid \mathbf{K} \cap \ell] = \Pr[\mathbf{x} \mid \mathbf{T}(\tau) \cap \ell]$ which allows us to calculate (1).

We classify problem instances according to properties of the mixture and the sampling process (i.e. the values of $r$, $m$, and $\tau$) into different regimes that necessitate different approaches to learn

the soft clustering. Naturally, for shorter trails, we require more difference in the transition processes between the CTMCs of the mixture, to be able to discern the trails. For longer trails, we can get away with less difference per state. However, it is important that the difference in the transition process is reflected in the discretized mixture $\mathbf{T}(\tau)$. For instance, if $\tau$ is chosen close to the mixing time in the chain, it is only possible to differentiate trails if the stationary distributions are distinct. We introduce the following learning regimes categorized by different trail lengths $m$.

### 3.2.1 Short to Medium Length.
For trails that are short such as those of length three or of a fixed (i.e., constant) length, or of medium length where $\tau m \ll t_{\mathrm{mix}}(\mathbf{K})$, encountering a state with notably distinct transition probabilities across different chains is crucial. This is vital for effectively differentiating trails from various chains. Such a state is referred to as a model difference. As the trail length transitions from short to medium, we anticipate an improvement in the quality of the soft assignment.

For $m = 3$ (the shortest length that allows learning a mixture [24]), we require a model difference in every state and non-zero starting probabilities to observe transitions from each state. In this case, we can use singular value decomposition (SVD) based algorithms [24, 40] to learn a mixture from the discretized trails $\mathbf{X}$, that aims to recover the discretized mixture $\mathbf{T}(\tau)$ via an estimate $\hat{\mathbf{T}}(\tau)$. From there, we derive the soft assignment by setting

$$\hat{a}(\mathbf{x}, \ell) := \frac{\Pr[\mathbf{x} \mid \hat{\mathbf{T}}(\tau) \cap \ell]}{\sum_{\ell'} \Pr[\mathbf{x} \mid \hat{\mathbf{T}}(\tau) \cap \ell']}. \qquad (2)$$

The following theorem establishes the recovery guarantee, and is based on the complex algebraic conditions that outline the model difference as detailed in [40].

**Theorem 2.** *If the discretized mixture $\mathbf{T}(\tau)$ fulfills the conditions of Theorem 1 in [40], we can recover $\mathbf{T}(\tau)$ and therefore obtain a valid $\hat{a}(\mathbf{x}, \ell)$ from the 3-trail distribution $p_{xyz} = \Pr[x_0 = x \wedge x_\tau = y \wedge x_{2\tau} = z]$ for all triples of states $x, y, z \in [n]$.*

The algorithm of [40] requires time $O(n^5 + n^3 L^3 + L^{\mathrm{cc}})$ where cc is the number of connected components in the mixture. In practice, we do not have access to the exact 3-trail distribution, but can estimate it from the transitions. Utilizing Chernoff bounds, it can be demonstrated that $O(n^3 \log n/\epsilon^2)$ transitions suffice to estimate this distribution up to $\pm\epsilon$ [24]. For trails of length $m > 3$, we use expectation maximization to learn an estimate $\hat{\mathbf{T}}(\tau)$. As in the case $m = 3$, we obtain a soft assignment from (2). This works well in practice especially when the number of transitions is low, but is merely a heuristic as convergence guarantees of expectation for mixtures of Markov chains are not known.

### 3.2.2 Long Length.
If trails are sufficiently long, we are able cluster them directly as in [28]. Intuitively, if $\tau m \gg t_{\mathrm{mix}}(\mathbf{K})$ and if the stationary distributions are all different, we are able to cluster the trails just by counting the number of visits to each state. Formally, let $\alpha$ and $\Delta$ be such that for all pairs of distinct chains $K, K' \in \{K^1, \dots, K^L\}$ there exists a state $y$ such that $\pi_K(y), \pi_{K'}(y) \ge \alpha$ and $\|K_y - K'_y\|_2 \ge \frac{1}{\tau}\Delta + 8\tau(1 + K_{\max}^2)$. That is, the state $y$ is visited sufficiently often and witnesses a model difference. We use the algorithm of [28] to obtain a clustering of the trails which we directly use for the assignment $\hat{a}(\mathbf{x}, \ell)$. We note that the obtained

clustering is hard, due to the long length of the trails. By Lemma 3 in Appendix C and [28, Theorem 1], we obtain the following result stated as a theorem:

THEOREM 3. *If we have* $r = \Omega(n^2 L^2 / \text{poly}(\Delta, \alpha))$ *trails of length*

$$m = \Omega\left(L^{1.5} t_{\text{mix}} \frac{\text{polylog}(r)}{\text{poly}(\Delta, \alpha)}\right)$$

*we obtain a valid soft clustering* $\hat{a}(\mathbf{x}, \ell)$ *with high probability.*

Using the method of [28] requires $O(n^3 + r^2 n^2)$ time.

*3.2.3    Very Long Length.* We are able to learn the discretized chain $T(\tau)$ from only a single trail in $\mathbf{X}^\ell$ for any $\ell \in [L]$ and we can obtain $\hat{a}(\mathbf{x}, \ell)$ as in (2), if the trail is long enough. The following theorem establishes the length of such a trail subject to $t_{\text{mix}}(\mathbf{K})$, the maximum mixing time in any chain, and $\pi_{\min} \coloneqq \min_{\ell, y} \pi_{K^\ell}(y)$.

THEOREM 4. *If* $\sum_{y=1}^n s_i^\ell = \Omega(1/L)$ *and we have* $r = \Omega(L \log L)$ *trails of length*

$$m = \Omega\left(\frac{1}{\pi_{\min}}\left(\frac{n}{\epsilon^2} + \frac{t_{\text{mix}}(\mathbf{K})}{\tau}\right)\log\frac{n}{\pi_{\min}}\right),$$

*then we can learn* $\mathbf{T}(\tau)$ *with recovery error at most* $\epsilon$ *with high probability.*

## 3.3    Recovery

We now show how to recover the individual chains in the mixture, given the discretized trails $\mathbf{X}$ and the valid soft clustering $\hat{a}(\mathbf{x}, \ell)$ which—depending on the chosen method in the previous section—is equal to or approximates $a(\mathbf{x}, \ell)$ up to permutation of $\ell$. We approach this via a Maximum Likelihood Estimation (MLE) given $a(\mathbf{x}, \ell)$, which means we want to find $\tilde{\mathbf{K}} = (\tilde{K}^1, \dots, \tilde{K}^L)$ to maximize the likelihood of observing $\mathbf{X}$ under knowledge of the posteriors $\Pr[\mathbf{x} \mid \mathbf{K} \cap \ell]$ for each $\mathbf{x} \in \mathbf{X}$. By the law of total probability, we can compute the probability of generating $\mathbf{X}$ from $\tilde{\mathbf{K}}$ as

$$\Pr[\mathbf{X} \mid \tilde{\mathbf{K}}] = \prod_{\mathbf{x} \in \mathbf{X}} \Pr[\mathbf{x} \mid \tilde{\mathbf{K}}] = \prod_{\mathbf{x} \in \mathbf{X}} \sum_{\ell=1}^L \Pr[\ell] \cdot \Pr[\mathbf{x} \mid \tilde{\mathbf{K}} \cap \ell] \tag{3}$$

To use the soft clustering and our approximate knowledge of the posterior $\Pr[\mathbf{x} \mid \mathbf{K} \cap \ell]$, we try to maximize the correlation instead of (3):

$$\prod_{\mathbf{x} \in \mathbf{X}} \sum_{\ell=1}^L \Pr[\ell] \cdot \Pr[\mathbf{x} \mid \tilde{\mathbf{K}} \cap \ell] \cdot \underbrace{\frac{\Pr[\mathbf{x} \mid \mathbf{K} \cap \ell]}{\sum_{\ell'} \Pr[\mathbf{x} \mid \mathbf{K} \cap \ell']}}_{=a(\mathbf{x}, \ell)} \tag{4}$$

The mixture $\tilde{\mathbf{K}}$ found by maximizing (4) serves as an approximation to the maximizer of (3), whose quality improves with the certainty of the soft clustering $a(\mathbf{x}, \ell)$. However, even maximizing (4) is difficult as we cannot optimize chains individually but have to consider their effect on the sample probability of each $\mathbf{x}$. Thus, instead of maximizing the arithmetic mean $\sum_{\ell=1}^L \Pr[\ell] \cdot \Pr[\mathbf{x} \mid \tilde{\mathbf{K}} \cap \ell] \cdot a(\mathbf{x}, \ell)$, we consider the geometric mean[1]

$$\prod_{\ell=1}^L \Pr[\ell] \cdot \Pr[\mathbf{x} \mid \tilde{\mathbf{K}} \cap \ell]^{a(\mathbf{x}, \ell)}.$$

---

[1]To shed light on the technical nuances, envision randomly picking from a mixture of two coins, each having success probabilities of $p$ and $q$ respectively. The resulting success probability becomes $\frac{p+q}{2}$. Rather than maximizing the true likelihood, we maximize $p^{1/2}q^{1/2}$, which serves as a lower limit.

Using this approximation, we can rewrite (4) as

$$\prod_{\mathbf{x} \in \mathbf{X}} \prod_{\ell=1}^L \Pr[\ell] \cdot \Pr[\mathbf{x} \mid \tilde{\mathbf{K}} \cap \ell]^{a(\mathbf{x}, \ell)}$$
$$= \prod_{\ell=1}^L \Pr[\ell]^{|\mathbf{X}|} \prod_{\mathbf{x} \in \mathbf{X}} \Pr[\mathbf{x} \mid \tilde{\mathbf{K}} \cap \ell]^{a(\mathbf{x}, \ell)} \tag{5}$$

In particular, we can now optimize each chain $\ell$ individually by maximizing the corresponding term in the RHS of (5). It remains to show that (5) is a good approximation for (4). Clearly, when $a(\mathbf{x}, \ell) \to 1$ for some chain $\ell$, the two terms also approach equality. However, we can even show that the terms are close when the entropy of $a(\mathbf{x}, \ell)$ for a fixed trail $\mathbf{x}$ is high:

THEOREM 5. *For each* $\mathbf{x} \in \mathbf{X}$,

$$\prod_{\ell=1}^L \Pr[\mathbf{x} \mid \mathbf{K} \cap \ell]^{a(\mathbf{x}, \ell)} \le \sum_{\ell=1}^L a(\mathbf{x}, \ell) \cdot \Pr[\mathbf{x} \mid \mathbf{K} \cap \ell]$$
$$\le L \cdot (\max_\ell a(\mathbf{x}, \ell)) \cdot \prod_{\ell=1}^L \Pr[\mathbf{x} \mid \mathbf{K} \cap \ell]^{a(\mathbf{x}, \ell)}.$$

Note that the above is tight whenever $a(\mathbf{x}, \ell)$ is uniform, over all chains $\ell \in [L]$. This shows that our approximation is good, for high and low entropy. We also establish the merit of this approximation experimentally in Section 4. Given the soft clustering, we can thus use an MLE to learn the individual chains and their starting probabilities. Specifically, we adapt the iterative heuristic introduced by [33] to use soft assignments. As an iterative heuristic, the MLE of [33] does not provide any convergence guarantees, but performs well in practice. The MLE step requires $O(n^3)$ time per iteration and per chain as well as scanning through each trail to pre-compute the transition counts $c_y^\ell$, which requires $O(rm)$ time.

## 3.4    Customizing the Algorithmic Framework

After presenting the three phases of our algorithmic framework for learning mixtures of CTMCs, we now describe three practical implementations that demonstrate both real-world efficiency, as discussed in Section 4, and adherence to the previously mentioned theoretical guarantees.

• GKV-ST: The SVD-based algorithm as referenced in [24, 40] is employed to learn mixtures of discrete-time Markov chains, leading to the soft clustering detailed in Section 3.2.1. Given that SVD-based techniques are tailored for discrete-time chains of length 3, we subdivide each discretized trail into segments of this length prior to the clustering phase.

• dEM: In lieu of the SVD-based algorithm, we use expectation maximization to learn a mixture of discrete-time Markov chains.

• KTT: We exclusively employ spectral clustering solely for the assignment step, as outlined in [28], with the underlying algorithm being credited to Vempala and Wang in their work [47]. This method is elaborated on in Section 3.2.2. It is noteworthy that this method ensures hard clustering, as per its algorithmic design.

## 4    EXPERIMENTAL EVALUATION

In our experimental analysis, we aim to answer the following key questions with experiments on synthetic data:

• What are the practical boundaries of the problem regimes, and how do different soft clusterings impact the performance of the algorithm? We investigate this in Figure 1 by varying the trail length.

• How accurate is the soft clustering and how much of the recovery error is attributed to error in the clustering? We examine

this by monitoring the clustering error over varying trail length $m$ in Figure 2. In Figure 8 in Appendix E, we show the recovery error while maintaining a constant number of transitions (i.e., $r \cdot m$ is constant); if we treat the number of transitions as a constant, the recovery error only depends on the error in the soft clustering.

• How much error is attributed to the recovery? Figure 7 in Appendix E the recovery error across various values of $\tau$.

Additional important experimental findings are shown in the Appendix E. We also apply our algorithms on two real-world scenarios, user trails on Last.fm and an NBA passing data set (public yet proprietary dataset obtained from Second Spectrum player tracking). Our results suggest that no single method is universally superior; however, dEM consistently performs well across various trail lengths, while KTT excels with extended trail lengths at a higher computational cost.

## 4.1 Experimental Setup

Due to space constraints, comprehensive information on our experimental setup can be found in Appendix E. All our findings are reproducible as the code is publicly available (see [1]).

**Synthetic Data.** We construct an underlying mixture of $L$ CTMCs as follows: for every chain $\ell \in [L]$, we randomly select a uniform rate matrix $K^\ell$ from the set of all rate matrices $K \in \mathbb{R}^{n \times n}$ with entries $K_{yz} \in [0, 1]$ for all distinct states $y, z$. Additionally, we randomly determine the starting probabilities $s^\ell \in \mathbb{R}^n$, drawing uniformly from the set of starting probabilities that sum up to 1 over $\ell \in [L]$ and $y \in [n]$. We sample $r$ continuous-time trails from the mixture according to the stochastic process described in Section 1. By monitoring the CTMC at consistent time intervals of $\tau$, we obtain discretized trails $\mathbf{x} = (x_{i\tau})_{0 \le i < m}$. In this process, only the first $m$ observations are retained.

**Last.fm dataset.** We obtain user trails from the Lastfm-1k dataset [2]. This dataset captures users' music listening history over three years, detailing each track played with associated user information, song title, and timestamp. For our study, we interpret a continuous sequence of songs listened to by a user as a single trail, provided that there are no interruptions exceeding 15 minutes. To streamline our data, we limit the states to the most frequently listened songs in the dataset and focus on users boasting the highest trail count. This results in a total of 2763 continuous-time trails with an average of 9 minutes of listening history. We discretize with $\tau = 10$ seconds.

**NBA dataset.** We use data from an exclusive NBA dataset from Second Spectrum to create continuous-time sequences for each team, where each player represents a state and state changes occur when players pass the ball. These trails capture offensive opportunities and conclude when the opposing team gains possession of the ball. We introduce two extra states, hit and miss to signify the success of each offense. The dataset refers to the 2022 and 2023 seasons, contains 1 433 788 passes made within 535 351 opportunities spanning 2 460 games and we generate 3850 sequences per team on average.

**Algorithms.** We use the three algorithms elucidated in Section 3.4: GKV-ST, KTT, and dEM. For dEM, we limit the discrete-time expectation maximization algorithm to a maximum of 100 iterations, typically ensuring adequate convergence. For synthetic data, we discretize with $\tau = 0.1$ unless otherwise specified.

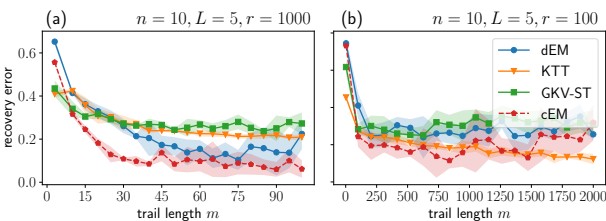

**Figure 1: Recovery error across different trail lengths: The plot illustrates two distinct scenarios: (a) A large number of transitions with shorter trails, and (b) a small number of transitions with long trails.**

In the second phase, when the objective is to recover a CTMC for a cluster of trails, we invariably opt for an approximation approach to the maximum likelihood estimator as proposed by [33] for a single chain. Given our CTMC mixture scenario's choice of soft cluster assignments, we have modified the method to cater for a weighted set of trails, as discussed in Section 3. For comparative analysis, we use continuous-time expectation maximization cEM that is given access to an initial span of $\tau \cdot m$ time for each continuous-time trail. We also attempted to train a mixture of CTMCs by employing the Python library HMMs on continuous-time hidden Markov models. Nevertheless, even with modest examples (such as when $L = 2, n = 5$), these attempts failed to reach a coherent mixture. Therefore, we omit these results from our presentation.

**Evaluation Metrics.** We evaluate the quality of the mixtures we have acquired using the recovery error, detailed in Section 1. To gauge the effectiveness of the soft clustering, we present the *clustering error* as:

$$\text{clustering-error}(a, a_{\text{gt}}) := \frac{1}{2|\mathbf{X}|} \min_{\sigma \in S_L} \sum_{\ell=1}^{L} \sum_{\mathbf{x} \in \mathbf{X}} |a(\mathbf{x}, \ell) - a_{\text{gt}}(\mathbf{x}, \sigma(\ell))|$$

In this context, $a$ is the soft clustering derived from our algorithms, while $a_{\text{gt}}$ signifies the ground truth. Specifically, $a_{\text{gt}}(\mathbf{x}, \ell) = 1$ if $\mathbf{x} \in \mathbf{X}^\ell$ and 0, otherwise.

**Machine specs.** We developed our software using Python 3 and executed it on a system powered by a 2.9 GHz Intel Xeon Gold 6226R processor, equipped with 384GB RAM.

## 4.2 Synthetic Experiments

To highlight the differences in our algorithms, we study situations with different trail lengths and numbers of transitions. In all experiments, we run each algorithm 5 times and report mean and standard deviation.

**Varying Trail Length and Number of Transitions.** Figure 1 presents the recovery error for our three proposed algorithms in scenarios with (a) abundant medium-length trails and (b) limited extended-length trails. In scenario (a), dEM mirrors the performance of cEM, achieving minimal recovery error given adequate samples. Conversely, scenario (b) highlights KTT's superior performance to dEM with extended trails, attributed to its enhanced clustering capabilities. However, as trails lengthen, calculating expectations becomes less stable. Notably, GKV-ST is restricted to trails of length 3, yet excels when supplied with numerous transitions. This trend is

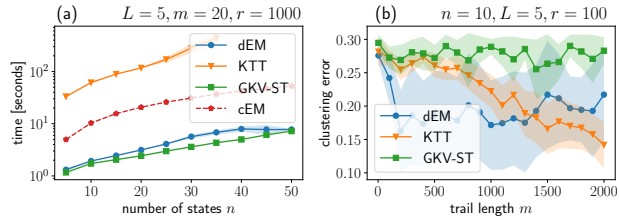

**Figure 2: Runtime in $n$ (a) and the clustering error (b).**

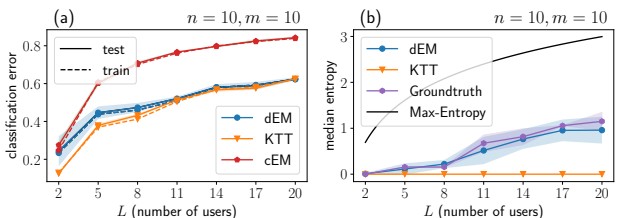

**Figure 3: Classification error and assignment entropy on the Last.fm dataset.**

further validated in Figures 8 and 9 in Appendix E, by maintaining a consistent total number of observations at (a) 2500 and (b) 25000 and contrasting medium and long trail scenarios respectively.

**Scalability.** Figure 2 (a) shows the scalability of our algorithms and the cEM baseline. We observe that KTT scales poorly in $n$. On the other hand, GKV-ST and dEM are much faster in practice (even though the former scales with $n^5$) and clearly outperform cEM. We showcase the scalability with respect to $L$ and $m$ in Appendix E.

**Clustering Error.** We generate a random mixture comprised of $L = 2$ chains and $n = 10$ states with $\tau = 0.1$. In Figure 2 (b), we plot the clustering error as the trail length varies. We observe that KTT is able to enhance the clustering quality as the length of the trails increases. However, dEM's clustering does not benefit from the use of trails longer than 200. We also observe that the variance of dEM is larger than KTT.

Overall, we conclude that our methods show differing behavior across problem regimes depending on $m$, $r$, but also in terms of scalability. Method selection is therefore problem specific.

### 4.3 Real-world Experiments

**Last.fm.** We select $k$ users from the *last.fm* dataset who have generated the most trails, where $k \in \{2, 5, 8, \ldots, 20\}$. We apply dEM, KTT and cEM by setting the chain count $L$ equal to $k$. We were not able to apply GKV-ST as the conditions of [40] are violated. This configuration inherently sets up a classification task, which is to classify the trails based on the originating user. Figure 3(a) plots the average classification error for both the train (depicted by dashed lines) and test dataset calculated across five 80%-20% train-test splits of the entire dataset. Using the hard clustering of KTT we obtain the best possible classification error. We also observe that this can be attributed to the good performance of the clustering step. Figure 3(b) shows the median entropy of the assignment of each trail to the chains. We observe that KTT even on the test data performs a low-entropy assignment, suggesting a near hard clustering. Interestingly, dEM produces an assignment that tracks the entropy of

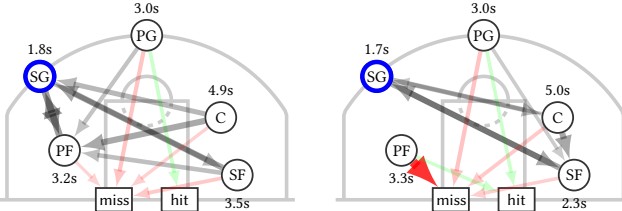

**Figure 4: Two Golden State Warriors offensive strategies from a mixture of $L = 4$ CTMCs, discretized over $\tau = 0.1$ seconds.**

the groundtruth. As the number of users in the dataset increases, the performance deteriorates due to the soft-assignment having to decide among a greater number of potential chains, coupled with the reality that some users have similar listening habits. In Appendix E we provide additional experiments, where a large number of users are represented by only a few chains, modeling a set of archetypical user patterns.

**Markovletics: Navigating Sports Strategies through CTMCs.** For every team in the 2022 season, we utilize trails crafted from an NBA passing dataset to deduce a blend of CTMCs. We aim for each chain to represent an offensive team's tactic, shedding light on the probability of point-scoring associated with that particular strategy by inspecting its steady-state distribution. For a qualitative evaluation of the learned mixtures, we asses its accuracy when used for a prediction of the score of an opportunity. We include this experiment along a detailed description of the setup in Section E. In Figure 4, we highlight two of the offensive tactics of Golden State Warriors (GSW) discerned from dEM with four CTMCs. The basketball game involves five positions: Point Guard (PG), Shooting Guard (SG), Power Forward (PF), Center (C), and Small Forward (SF). Each position is annotated with the calculated ball holding time. Arrows represent potential passes between positions, with their thickness and opacity denoting the pass's probability. Passes with a low likelihood (less than 0.2) are excluded for visual convenience. The player with the highest starting probability is highlighted in blue. Shoot attempts are highlighted in red (miss) and green (hit). Each strategy illuminates unique offensive patterns. We also note that the strategy on the left has a 44% chance of scoring, while the one on the right has a probability of 37%. Additional strategies from GSW can be found in Appendix E, alongside strategies of the New York Knicks.

## 5 CONCLUSION

This research delved extensively into the study of learning mixtures of CTMCs, presenting novel algorithms and conducting comparisons with leading competitors across synthetic and real-world scenarios. Our methods have been proven effective in real-world scenarios, as seen in the *Last.fm* application. Additionally, we introduced the innovative concept of *Markovletics* for learning offensive tactics in NBA. In essence, our research adds a fresh dimension to the theoretical aspects of Markov chains and exemplifies its real-world applicability. Our study raises several intriguing questions, including the choice of the parameter $L$ and the expansion of our techniques to a broader spectrum of datasets, including those from bioinformatics [7].

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

## A EFFECT OF THE $\tau$ PARAMETER

We remind the reader of the following formal definition of a CTMC with rate matrix $K \in \mathbb{R}^{n \times n}$. Let $M \in \mathbb{R}^{n \times n}$ be the transition matrix of a discrete-time Markov chain given by $M_{yz} := K_{xy}/|K_{yy}|$ for $y \neq z$ and $M_{yy} = 0$. Let $Y^d(i) \in [n]$ be the state of a random walk through $M$ at step $i \in \mathbb{N}_0$. To define the continuous-time process, we sample transition times $T(i) \sim \text{Exp}(|K_{yy}|)$ where $y = Y^d(i)$. Now, for a time $t \geq 0$, we set $Y^c(t) = Y^d(i)$ where $i \in \mathbb{N}_0$ is such that $\sum_{j<i} T(j) \leq t < \sum_{j\leq i} T(j)$.

In the remainder of this section, we provide further intuition on the quality of estimation of $\mathbf{K}$ for different values of $\tau$, as discussed in the introduction and Section 3.1.

Figure 5 demonstrates the significance of selecting an appropriate value for $\tau$. The illustration depicts transitions from state 1 to state 2 in two separate chains, each with distinct transition rates of 1 and 10, respectively. The choice of the time scale $\tau$ has a substantial impact on the observed transition probabilities. It is essential to carefully choose $\tau$ (e.g., $\tau = 0.1$) in order to distinguish between the discretized chains, as we discussed in Section 3.1. The underlying issue is that when one chain within the mixture transitions significantly faster than another, using the correct discretization choice becomes critical. A too-small discretization may result in too few observed transitions in the slower chain, while a too-large discretization may cause both chains to converge to their potentially identical stationary distributions, hindering the differentiation of the trajectories.

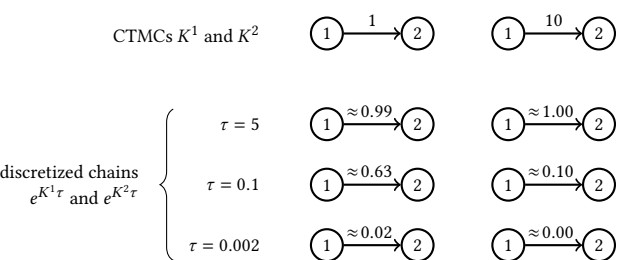

**Figure 5: Choosing the right value of $\tau$ is crucial.**

Figure 6 shows the asymptotic trends of the variables in the estimation of $\mathbf{K}$ with respect to $\tau$ that we introduced in Section 3.1: the total observation time for holdings periods without transitions, the count of observed transitions and the tally of bad transitions. Owing to the varying gradient as $\tau$ approaches 0, we can adjust $\tau$ to yield a minimal set of bad transitions while ensuring a significant number of quality (i.e., not bad) transitions.

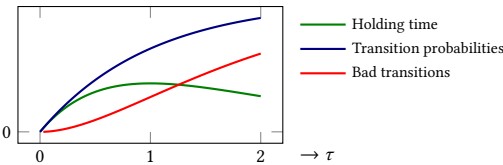

**Figure 6: Estimating K.**

## B PROOFS FROM SECTION 3.1

We define $q_y^\ell := e^{K_{yy}^\ell \tau} = \Pr[E \geq \tau]$ for $E \sim \text{Exp}(|K_{yy}|)$ as the probability to remain in state $y$ in time $\tau$. Since we are observing states at discrete time intervals of size $\tau$, our estimator is

$$\hat{q}_y^\ell = \frac{1}{c_y^\ell} \left| \{\mathbf{y} \in \mathbf{X}^\ell, 0 \leq i < m : \mathbf{x}_i = y \wedge \mathbf{x}_{i+1} = y\} \right|$$

where $c_y^\ell = \left| \{\mathbf{x} \in \mathbf{X}^\ell, i : \mathbf{x}_i = y\} \right|$ is the number of transitions from chain $\ell$ that traverse through $y$. Note that $\hat{q}_y^\ell$ is a biased estimator of $q_y^\ell$ because we are unable to tell whether transitions are bad (see Definition 1). That is, when estimating the holding time with our estimator, it could happen that $\mathbf{x}_i = x_{\tau i} = y$ and $\mathbf{x}_{i+1} = x_{\tau(i+1)} = y$ but there is a $\xi \in (0,1)$ with $x_{\tau i+\xi} \neq y$. We thus have to set $\tau$ small enough to avoid bad samples. In particular, we set

$$\tau = \frac{\sqrt{\epsilon_h}}{3K_{\max}}.$$

We now estimate one row of the rate matrix $K_y^\ell = (K_{yz}^\ell)_z$ for a fixed chain $\ell$ and state $y$. For brevity, we will omit $\ell$ in $K_{yz} = K_{yz}^\ell$ and omit $z$ in $q = q_z^\ell$, $\hat{q} = \hat{q}_y^\ell$, and $m = m_y^\ell$.

LEMMA 1. *Given $c = \Omega\left(e^{|K_{yy}|\tau} \epsilon_h^{-2} \log(Ln)\right) = \Omega\left(\epsilon_h^{-2} \log(Ln)\right)$ transitions, our estimator $\hat{q}$ for the holding time satisfies*

$$|\hat{q} - q| \leq \epsilon_h q$$

*with high probability.*

PROOF. We first bound the number of bad transitions. In particular, we want that only a $2\gamma := \frac{\epsilon_h}{2} q$ fraction of the $c$ transitions are bad so that we do not incur to much error from these samples. We first calculate the probability to obtain a single bad sample. Here, we use that the larger $|K_{yy}|$, the more likely it is to switch states, so the probability of obtaining a bad sample is maximized in the state with largest $|K_{yy}|$. Let thus $E, E' \sim \text{Exp}(K_{\max})$ be independent random variables. By the memorylessness of the exponential distribution,

$$\Pr[E + E' < \tau] = \int_0^\tau \Pr[E' \leq \tau - t] f_E(t) dt$$
$$= K_{\max} \int_0^\tau \left(1 - e^{-K_{\max}(\tau-t)}\right) e^{-K_{\max}t} dt$$
$$= 1 - (1 + K_{\max}\tau) e^{-K_{\max}\tau}$$
$$\leq 1 - (1 + K_{\max}\tau)(1 - K_{\max}\tau)$$
$$= K_{\max}^2 \tau^2$$

where $f_E$ is the PDF of $E$. We further bound

$$K_{\max}^2 \tau^2 = \frac{\epsilon_h}{9} \leq \frac{\epsilon_h}{4} - \frac{\epsilon_h}{12} \cdot \frac{\sqrt{\epsilon_h} |K_{xx}|}{K_{\max}} \leq \frac{\epsilon_h}{4} \left(1 - \frac{1}{3}\sqrt{\epsilon_h}\frac{|K_{xx}|}{K_{\max}}\right)$$
$$\leq \frac{\epsilon_h}{4} e^{\frac{1}{3}\sqrt{\epsilon_h}\frac{K_{xx}}{K_{\max}}} = \frac{\epsilon_h}{4} e^{K_{xx}\tau} = \frac{\epsilon_h}{4} q = \gamma.$$

Let now $H_j = 1$ if there is no state change in the observed transition $j \in [c]$. Let $B_j = 1$ if the $j$-th transition is bad. By a Chernoff bound,

$$\Pr\left[\sum_{j=1}^c B_j \geq 2c\gamma\right] \leq \exp\left(-\frac{1}{3}c\gamma\right)$$
$$= \exp\left(-\frac{1}{12}\epsilon_h c q\right) = O\left(\frac{1}{\text{poly}(Ln)}\right).$$

Thus, with high probability, at most a $2\gamma$ fraction of the samples is bad. We therefore incur an additive error of at most $2\gamma = \frac{\epsilon_{\mathrm{h}}}{2}q$ from bad samples. Since $2\gamma \leq \frac{1}{2}$, we can use another Chernoff bound on the good samples, which make up at least half the total samples, to show that the estimation error from sampling is at most $\frac{\epsilon_{\mathrm{h}}}{2}$ with high probability:

$$\Pr\left[\left|q - \frac{2}{c}\sum_{j=1}^{c/2}H_j\right| \geq \frac{\epsilon_{\mathrm{h}}}{2}q\right] \leq 2\exp\left(-\frac{1}{24}\epsilon_{\mathrm{h}}^2 cq\right)$$
$$= O\left(\frac{1}{\mathrm{poly}\,(Ln)}\right). \quad \square$$

Let $p_z := \frac{K_{yz}}{|K_{yy}|}$ be the transition probability from state $y$ to $z$ within time $\tau$. We estimate $p_z$ through

$$\hat{p}_z := \frac{\left|\left\{\mathbf{x}\in\mathbf{X}^\ell, 0\leq i < m : \mathbf{x}_i = y \wedge \mathbf{x}_{i+1} = z\right\}\right|}{\left|\left\{\mathbf{x}\in\mathbf{X}^\ell, 0\leq i < m : \mathbf{x}_i = y \wedge \mathbf{x}_{i+1} \neq y\right\}\right|}$$

for all states $y \neq x$. Let also $\mathbf{p} := (p_z)_{z\neq y}$ and $\hat{\mathbf{p}} := (\hat{p}_z)_{z\neq y}$ be discrete probability vectors.

LEMMA 2. With $c = \Omega\left(\frac{n\kappa}{\epsilon_{\mathrm{t}}^2\sqrt{\epsilon_{\mathrm{h}}}}\log(Ln)\right)$ transitions, our estimator for the transition probabilities satisfies

$$\mathrm{TV}(\mathbf{p}, \hat{\mathbf{p}}) \leq \epsilon_{\mathrm{t}}$$

with high probability.

PROOF. We first bound the probability to see a transition in a given sample. Since the probability of observing a transition is minimized for $K_{\min}$, let $E = \mathrm{Exp}(K_{\min})$. Using the fact that $e^{-x} \geq 1 - \frac{x}{2}$ for $x \in [0, 1.59]$, we can bound the probability of observing a transition as

$$\Pr[E < \tau] = 1 - e^{-K_{\min}\tau} = 1 - e^{-\kappa^{-1}\frac{\sqrt{\epsilon_{\mathrm{h}}}}{3}} \geq \frac{\sqrt{\epsilon_{\mathrm{h}}}}{6\kappa}.$$

We obtain that at least a $\frac{\sqrt{\epsilon_{\mathrm{h}}}}{12\kappa}$ fraction of the samples are transitions with high probability, since by a Chernoff bound,

$$\Pr\left[\frac{1}{m}\sum_{i=1}^c T_j \leq \frac{\sqrt{\epsilon_{\mathrm{h}}}}{12\kappa}\right] = \Pr\left[\sum_{i=1}^c T_j \leq \left(1 - \frac{1}{2}\right)c\frac{\sqrt{\epsilon_{\mathrm{h}}}}{6\kappa}\right]$$
$$\leq \exp\left(-c\frac{\sqrt{\epsilon_{\mathrm{h}}}}{\kappa}\cdot\frac{1}{48}\right) = O\left(\frac{1}{\mathrm{poly}\,(Ln)}\right).$$

where $T_j = 1 - H_j$ is a random variable for observing a state change in the $j$-th transition. With a similar argument as in Lemma 1, we can show that if $\kappa\sqrt{\epsilon_{\mathrm{h}}} \leq \frac{9}{24}\epsilon_{\mathrm{t}}$, only a $\frac{\epsilon_{\mathrm{t}}}{2}$-fraction of the transition samples are bad. Furthermore, it is well know that to estimate a discrete distribution with support $n - 1$ to $\mathrm{TV}(\mathbf{p}, \hat{\mathbf{p}}) \leq \frac{\epsilon_{\mathrm{t}}}{2}$, we require $\Omega(n/\epsilon_{\mathrm{t}}^2)$ many samples of transitions. This requires that $\frac{\sqrt{\epsilon_{\mathrm{h}}}}{12\kappa}c = \Omega(n/\epsilon_{\mathrm{t}}^2)$ which is satisfied by setting $c$ as in the theorem statement. $\square$

Since $e^{K_{yy}\tau} = q$, we set

$$\hat{K}_{yy} := \frac{\log(\hat{q})}{\tau} \qquad \text{and} \qquad \hat{K}_{yz} := \hat{p}_z\left|\hat{K}_{yy}\right|.$$

We define the $y$-th row of the $\ell$-th chain as $K_y^\ell = (K_{yz})_z$ and our estimate as $\hat{K}_y^\ell = (\hat{K}_{yz})_z$.

Theorem 1 *Using* $c = \Omega\left(\frac{\kappa^2}{\epsilon^3}\left(n + \frac{\kappa^2}{\epsilon}\right)\log(Ln)\right)$ *transitions, we can estimate* $K_y^\ell$ *such that* $\mathrm{TV}(K_y^\ell, \hat{K}_y^\ell) \leq \epsilon$ *with high probability.*

PROOF. By the definition of the TV distance,

$$\mathrm{TV}(K_y^\ell, \hat{K}_y^\ell) = \frac{1}{2}\int_0^\infty \sum_{z\neq y}\left|\frac{\hat{K}_{yz}}{\hat{K}_{yy}}\hat{K}_{yy}e^{\hat{K}_{yy}t} - \frac{K_{yz}}{K_{yy}}K_{yy}e^{K_{yy}t}\right|dt$$
$$= \frac{1}{2\tau}\int_0^\infty \sum_{z\neq y}\left|\hat{p}_z\log(\hat{q})\,\hat{q}^{t/\tau} - p_z\log(q)\,q^{t/\tau}\right|dt.$$

We condition on the case that $|\hat{q} - q| \leq \epsilon_{\mathrm{h}}q$ and $\mathrm{TV}(\mathbf{p}, \hat{\mathbf{p}}) \leq \epsilon_{\mathrm{t}}$ which both happen with high probability due to Lemma 1 and Lemma 2, respectively. We can then apply the bound

$$\left|\hat{a}\hat{b} - ab\right| = \left|\hat{a}(\hat{b} - b) + (\hat{a} - a)b\right| \leq |\hat{b} - b|\cdot|\hat{a}| + |\hat{a} - a|\cdot|b| \quad (6)$$

twice to each inner term. That is, we first bound

$$\left|\hat{q}^{t/\tau}\log(\hat{q}) - q^{t/\tau}\log(q)\right|$$
$$\leq \left|\log(\hat{q}) - \log(q)\right|\cdot\hat{q}^{t/\tau} + \left|\hat{q}^{t/\tau} - q^{t/\tau}\right|\cdot|\log(q)|$$
$$\leq 2\epsilon\hat{q}^{t/\tau} + \left|\hat{q}^{t/\tau} - q^{t/\tau}\right|\cdot|\log(q)|.$$

since $|\log(\hat{q}) - \log(q)| = \log\left(\max\left\{\frac{\hat{q}}{q}, \frac{q}{\hat{q}}\right\}\right) \leq e^{2\epsilon_{\mathrm{h}}}$. We use this to bound

$$\left|\hat{p}_z\log(\hat{q})\,\hat{q}^{t/\tau} - p_z\log(q)\,q^{t/\tau}\right|$$
$$\leq \left|\log(\hat{q})\,\hat{q}^{t/\tau} - \log(q)\,q^{t/\tau}\right|\cdot\hat{p}_z + |\hat{p}_z - p_z|\cdot\left|\log(q)\,q^{t/\tau}\right|$$
$$\leq 2\epsilon_{\mathrm{h}}\hat{q}^{t/\tau}\hat{p}_z + \left|\hat{q}^{t/\tau} - q^{t/\tau}\right|\cdot|\log(q)|\cdot\hat{p}_z + |\hat{p}_z - p_z|\cdot\left|\log(q)\,q^{t/\tau}\right|.$$

Plugging this back in, we obtain

$$\mathrm{TV}(K_y^\ell, \hat{K}_y^\ell) \leq \frac{1}{2}\int_0^\infty \sum_{z\neq y}\Bigg(\underbrace{2\frac{\epsilon_{\mathrm{h}}}{\tau}\hat{q}^{t/\tau}\hat{p}_z}_{(I)}$$
$$+ \underbrace{\left|\hat{q}^{t/\tau} - q^{t/\tau}\right|\cdot\frac{|\log(q)|}{\tau}\cdot\hat{p}_z}_{(II)} + \underbrace{|\hat{p}_z - p_z|\cdot\left|\frac{\log(q)}{\tau}q^{t/\tau}\right|}_{(III)}\Bigg)dt.$$

We analyze all three error terms separately. First, we bound

$$(I) = 2\frac{\epsilon_{\mathrm{h}}}{\tau}\sum_{z\neq y}\hat{p}_{yz}\int_0^\infty \hat{q}^{t/\tau}dt = 2\epsilon_{\mathrm{h}}\frac{1}{|\log(\hat{q})|}$$
$$\leq 2\epsilon_{\mathrm{h}}\frac{1}{|\log(q)| - \log(1 + \epsilon_{\mathrm{h}})} = 2\frac{\epsilon_{\mathrm{h}}}{|K_{yy}|\tau - \epsilon_{\mathrm{h}}}$$

and, assuming that $\sqrt{\epsilon_{\mathrm{h}}} \leq \kappa^{-1}$,

$$(II) = \int_0^\infty \sum_{z\neq y}\left|\hat{q}^{t/\tau} - q^{t/\tau}\right|\cdot\frac{|\log(q)|}{\tau}\cdot\hat{p}_zdt$$
$$= |K_{yy}|\cdot\int_0^\infty\left|\hat{q}^{t/\tau} - q^{t/\tau}\right|dt$$
$$\leq |K_{yy}|\left(\frac{1}{|K_{yy}| - \frac{\epsilon_{\mathrm{h}}}{\tau}} - \frac{1}{|K_{yy}|}\right)$$
$$= \frac{\epsilon_{\mathrm{h}}}{|K_{yy}|\tau - \epsilon_{\mathrm{h}}}.$$

Finally,

$$(\text{III}) = \int_0^\infty \sum_{z \neq y} |\hat{p}_z - p_z| \cdot \left| \frac{\log(q)}{\tau} q^{t/\tau} \right| dt$$

$$= \sum_{z \neq y} |\hat{p}_z - p_z| \cdot |K_{yy}| \int_0^\infty e^{K_{yy}t} dt$$

$$= \sum_{z \neq y} |\hat{p}_z - p_z|$$

$$= 2\text{TV}(\mathbf{p}, \hat{\mathbf{p}}).$$

Thus, we incur a total error of

$$\text{TV}(K_y^\ell, \hat{K}_y^\ell) \leq 2\frac{\epsilon_\text{h}}{|K_{yy}|\,\tau - \epsilon_\text{h}} + \text{TV}(\mathbf{p}, \hat{\mathbf{p}}) \leq 12\kappa\sqrt{\epsilon_\text{h}} + \epsilon_\text{t}$$

since $|K_{yy}|\,\tau - \epsilon_\text{h} \geq \frac{1}{6}\sqrt{\epsilon_\text{h}}\kappa^{-1}$ if $6\sqrt{\epsilon_\text{h}} \leq \kappa^{-1}$. We can thus set $\sqrt{\epsilon_\text{h}} = \frac{\epsilon}{24\kappa}$ and $\epsilon_\text{t} = \frac{\epsilon}{2}$ to obtain $\text{TV}(K_y, \hat{K}_y) \leq \epsilon$. With Lemma 1 and Lemma 2, this gives us the bound on $m$ in the theorem statement. □

## C PROOFS FROM SECTION 3.2

**Long trails.** In order to apply the theorem of [28], we need to ensure that each pair of chains exhibits a model difference $\Delta$ in a state that is visited sufficiently often. To ensure the former, we show how to transfer a model difference from the mixture of CTMCs to a mixture of discrete-time Markov chains:

LEMMA 3. *For any two rate matrices* $K, K' \in \{K^1, \dots, K^L\}$ *and a state* $y \in [n]$, *if* $\|K_y - K'_y\|_2 \geq \frac{1}{\tau}\Delta + 8\tau(1 + K_{\max}^2)$ *then* $\|e_y^{K\tau} - e_y^{K'\tau}\|_2 \geq \Delta$.

PROOF. By the definition of the matrix exponential as a Taylor series,

$$e^{K\tau} - e^{K'\tau} = \tau(K - K') - \sum_{k=2}^\infty \frac{1}{k!}((K\tau)^k - (K'\tau)^k)$$

and thus, by the triangle inequality,

$$\|e_y^{K\tau} - e_y^{K'\tau}\|_2 \geq \tau\|K_y - K'_y\|_2 - \|A_y\|_2 - \|A'_y\|_2 \quad (7)$$

where $A := \sum_{k=2}^\infty (K\tau)^k$ and $A'$ is defined analogously. It thus remains to analyze $\|A_y\|$. To this end, let again $K_{\max} := \max_{\ell,y} |K_{yy}^\ell|$. Since rate matrices are diagonally dominant, we know by the Gershgorin circle theorem that all eigenvalues of $K\tau$ lie in $[-2K_{\max}\tau, 0]$. It is easy to see that the magnitude of the eigenvalues of $A$ are thus bounded by

$$|\lambda(A)| \leq \sum_{k=2}^\infty \frac{1}{k!}(2K_{\max}\tau)^k = e^{2K_{\max}\tau} - 1 - 2K_{\max}\tau \leq 4K_{\max}^2\tau^2$$

for $K_{\max}\tau \leq \frac{1}{2}$ and therefore $\|A_y\|, \|A'_y\| \leq 4K_{\max}^2\tau^2$. We plug this back into (7) and obtain $\|e_y^{K\tau} - e_y^{K'\tau}\|_2 \geq \tau\|K_y - K'_y\|_2 - 8K_{\max}^2\tau^2 \geq \Delta$. □

When setting $\tau^{-1} = \Theta(\kappa K_{\max})$ in the as in the context of Section 3.1, we obtain: If $\|K_y - K'_y\|_2 = \kappa\Omega(\kappa K_{\max}\Delta + K_{\min})$ then $\|e_y^{K\tau} - e_y^{K'\tau}\|_2 \geq \Delta$.

**Very Long Trails.** We define the $\pi$-norm of a vector $u \in \mathbb{R}^n$ through

$$\|u\|_\pi^2 = \sum_{i=1}^n \frac{u_i^2}{\pi(i)}.$$

THEOREM 6 (CHERNOFF-HOEFFDING BOUND FOR RANDOM WALKS [15]). *Let* $\mathbf{x} \in [n]^m$ *be a random walk of length* $m$ *in a discrete-time Markov chain* $M \in \mathbb{R}^{n \times n}$ *with associated starting probabilities* $s \in [0, 1]^n$. *Let the stationary distribution of* $M$ *be* $\pi$ *and the mixing time* $t_\text{mix}$. *Let* $f \colon [n] \to [0, 1]$ *be a function with* $\mu := \mathbb{E}_{y \sim \pi}[f(y)]$ *and* $F := \sum_{z \in \mathbf{x}} f(z)$. *Then,*

$$\Pr[F \geq (1 + \delta)\mu m] \leq \|s\|_\pi \cdot \begin{cases} e^{-\Omega(\delta^2 \mu m / t_\text{mix})} & \text{for } 0 \leq \delta \leq 1 \\ e^{-\Omega(\delta \mu m / t_\text{mix})} & \text{for } \delta \geq 1. \end{cases}$$

For some fixed state $y$, let $f(z) = 0$ if $z = y$ and $f(z) = 1$, otherwise, such that $c_y := \sum_{z \in \mathbf{x}} f(z) = m - F$ is the number of times a random walk $\mathbf{x}$ traverses state $y$. We bound the probability of obtaining less than $\theta$ samples:

$$\Pr[c_y \leq \theta] \leq \Pr[F \geq (1 + \delta)\mu m] \leq \|s\|_\pi e^{-\Omega(\delta\mu m/t_\text{mix})} \quad (8)$$

for $\mu = 1 - \pi(y)$ and $\delta = \Omega(\frac{\pi(y) - \theta/m}{\mu})$.

THEOREM 7. *Given a Markov chain with transition probabilities* $M$, *we can learn* $M$ *up to recovery error* $\epsilon$ *from a single trail of length*

$$m = \Omega\left(\frac{n\epsilon^{-2} + t_\text{mix}(M)}{\pi_\text{min}} \log \frac{n}{\pi_\text{min}}\right)$$

*with high probability.*

Let $M_y = (M_{yz}) \in \mathbb{R}^n$ be the vector of transition probabilities out of state $y$. We define an estimator $\hat{M}_y$ through $\hat{M}_{yz} := c_{yz}/c_y$, where $c_{yz}$ is the number of times the trail transitions from state $y$ to $z$ and $c_y := \sum_z c_{yz}$.

PROOF. We first bound the TV-distance from $M_y$ to $\hat{M}_y$ for a fixed count $c_y$. Choose an arbitrary set of states $Z \subseteq [n]$ and let $M_y(Z) := \sum_{z \in Z} M_{yz}$. By a Hoeffding bound,

$$\Pr\left[|M_y(Z) - \hat{M}_y(Z)| > \epsilon\right] \leq 2e^{-2c_y\epsilon^2}.$$

By a union bound, the probability that any set $Z$ experiences error more than $\epsilon$ under $\theta$ observations is at most

$$\Pr\left[\text{TV}(M_y, \hat{M}_y) > \epsilon\right] \leq 2^n \cdot 2e^{-2c_y\epsilon^2} \leq e^{n - 2c_y\epsilon^2} \quad (9)$$

for sufficiently large $n$. Let now

$$\theta = O\left(m\pi(y) + \frac{n - m\pi(y)\epsilon^2}{1/t_\text{mix} + \epsilon^2}\right)$$

where $t_\text{mix} := t_\text{mix}(M)$. We consider two bad events. First, that there are not enough samples, i.e. $c_y \leq \theta$. Second, that given at least $\theta$ samples, the estimation error is larger than $\epsilon$. Using (8), the probability of the first bad event is at most

$$\Pr[c_y \leq \theta] \leq \|s\|_\pi e^{-\Omega(\delta\mu m/t_\text{mix})} = \|s\|_\pi \exp\left(-\Omega\left(\frac{m\pi(y) - n\epsilon^{-2}}{\epsilon^{-2} + t_\text{mix}}\right)\right).$$

Furthermore, the probability of the second bad event is, due to (9),

$$\Pr[\text{TV}(M_y, \hat{M}_y) > \epsilon \mid c_y \geq \theta] \leq \exp\left(-\Omega\left(\frac{m\pi(y) - n\epsilon^{-2}}{\epsilon^{-2} + t_\text{mix}}\right)\right).$$

Combining both, we see that probability that of error $\geq \epsilon$ when estimating $M_x$ is at most

$$\Pr[\mathrm{TV}(M_y, \hat{M}_y) > \epsilon] \leq \|s\|_\pi \exp\left(-\Omega\left(\frac{m\pi(y) - n\epsilon^{-2}}{\epsilon^{-2} + t_{\mathrm{mix}}}\right)\right)$$

$$\leq \frac{1}{\sqrt{\pi_{\min}}} \exp\left(-\Omega\left(\frac{m\pi_{\min} - n\epsilon^{-2}}{\epsilon^{-2} + t_{\mathrm{mix}}}\right)\right).$$

By a union bound, the probability that for a fixed chain, any state has estimation error more than $\epsilon$ is at most

$$\frac{n}{\sqrt{\pi_{\min}}} \exp\left(-\Omega\left(\frac{m\pi_{\min} - n\epsilon^{-2}}{\epsilon^{-2} + t_{\mathrm{mix}}}\right)\right)$$

Therefore, with high probability, we are able to estimate the chain from a single trail within error $\epsilon$ when setting $m$ as the the theorem statement. $\qquad\square$

Finally, we remark that the probability that there is a chain for which we do not obtain a single trail is, by a union bound, at most

$$\sum_{\ell=1}^{L} (1 - \|s^\ell\|_1)^m = L \cdot (1 - \Omega(1/L))^m \leq L \cdot e^{-\Omega(m/L)}.$$

It thus suffices to set $m = \Omega(L \log L)$ to obtain samples from ever chain with high probability.

## D PROOFS FROM SECTION 3.3

THEOREM 5. *For each $\mathbf{x} \in \mathbf{X}$,*

$$\prod_{\ell=1}^{L} \Pr[\mathbf{x} \mid \mathbf{K} \cap \ell]^{a(\mathbf{x},\ell)} \leq \sum_{\ell=1}^{L} a(\mathbf{x}, \ell) \cdot \Pr[\mathbf{x} \mid \mathbf{K} \cap \ell]$$

$$\leq L \cdot (\max_\ell a(\mathbf{x}, \ell)) \cdot \prod_{\ell=1}^{L} \Pr[\mathbf{x} \mid \mathbf{K} \cap \ell]^{a(\mathbf{x},\ell)}.$$

PROOF (THEOREM 5). The first inequality can be directly derived from the arithmetic-geometric mean inequality using the factors $\hat{a}(\mathbf{x}, \ell)$. For the second, it is worth noting that without loss of generality we can assume $\sum_{\ell=1}^{L} \Pr[\mathbf{x} \mid \mathbf{K} \cap \ell] = 1$. This is because both the arithmetic and geometric mean scale linearly. We thus want to show that

$$\sum_{\ell=1}^{L} p_\ell^2 \leq C \cdot \prod_{\ell=1}^{L} p_\ell^{p_\ell} \tag{10}$$

for $p_\ell = \hat{a}(\mathbf{x}, \ell) = \Pr[\mathbf{x} \mid \mathbf{K} \cap \ell]$. Note that (10) is equivalent to

$$2^{\log_2 \sum_{\ell=1}^{L} p_\ell^2} = 2^{-H_2(\mathbf{p})} \leq C \cdot 2^{-H_1(\mathbf{p})} = C \cdot 2^{\sum_{\ell=1}^{L} p_\ell \log_2 p_\ell}$$

where $H_\alpha(\mathbf{p})$ is the Rényi entropy[2] of $\mathbf{p}$ and a suitable constant $C > 0$. By well-known facts about the Rényi-entropy,

$$H_1(\mathbf{p}) - H_2(\mathbf{p}) \leq \log_2 L - H_2(\mathbf{p})$$

$$\leq \log_2 L - H_\infty(\mathbf{p}) = \log_2 L + \log_2 \max_\ell p_\ell$$

so we obtain $C = L \cdot \max_\ell p_\ell$ as required for the theorem statement. $\qquad\square$

---

[2]The Rényi entropy of order $\alpha$ is defined as $H_\alpha(\mathbf{p}) = \frac{1}{1-\alpha} \log_2 \left(\sum_{\ell=1}^{L} p_i^\alpha\right)$ while the Shannon entropy $H_1(\mathbf{p})$ and min-entropy $H_\infty(\mathbf{p}) = -\log_2 \max_\ell p_\ell$ are defined trough the limit.

## E EXPERIMENTAL EVALUATION

### E.1 Detailed Dataset Description

**Last.fm dataset.** We obtain user trails from the Lastfm-1k dataset which can be accessed via the provided link [2]. This dataset captures users' music listening history over three years, detailing each track played with associated user information, song title, and timestamp. For our study, we interpret a continuous sequence of songs listened to by a user as a single trail, provided that there are no interruptions exceeding 15 minutes. To streamline our data, we limit the states to the 10 most frequently listened songs in the dataset and focus on users boasting the highest trail count. This results in a total of 2763 continuous-time trails with an average of 9 minutes of listening history. We then convert the trails using a time frame of $\tau = 10$ seconds. Following this, the converted discrete trails are segmented into smaller trails, each with a length of 10. We obtain 13 615 discrete-time trails.

**NBA Dataset.** The NBA dataset from Second Spectrum archives every pass executed during NBA basketball games for the 2022 and 2023 seasons. Each documented pass is linked with a specific offensive opportunity and is marked with the time it was made, as well as the passer and the receiver. In this context, an opportunity refers to a continuous duration when a team possesses the ball. This record also mentions the team on the offense and the points they score during the possession. It is important to note that in NBA rules (rule no 7), a team's opportunity to score is constrained to 24 seconds due to the shot clock regulation. This comprehensive dataset consists of 1 433 788 passes made within 535 351 opportunities spanning 2 460 games.

To dissect the data for each team during the two seasons, we employ the following approach. We designate a state for each of the top 12 players who have the ball for the longest durations. In addition, we introduce two special distinct states hit and miss. These states signify whether an opportunity culminated in the offensive team scoring or failing to score, respectively. The continuous-time trail begins at the state that corresponds to the player who receives the opening pass. The subsequent state is the receiving player of the next pass, and the transition time mirrors the time lapse between the two passes. When an opportunity wraps up, the trail concludes (i.e., is absorbed in terms of Markov chains) in the hit or miss state, contingent on the scoring outcome. Our analysis only includes opportunities that span beyond 5 seconds and involve a minimum of 3 passes. To ensure balance, we exclude any surplus of hit and miss opportunities. This leaves us with a total of 3850 trails per team on average.

### E.2 Additional Experimental Results

*Remark.* It is important to mention that in cases where the $y$-axis of a figure lacks annotation, it corresponds to the $y$-axis of the adjacent figure on the left.

**Varying $\tau$.** Figure 7 shows the effect of using different $\tau$ values on the recovery error. Note that dEM, KTT, and GKV−ST utilize $\tau$ as a discretization parameter. In contrast, cEM operates on trails generated by the CTMC, continually observed for a duration of $\tau \cdot m$. We can see that, as the observation duration increases, the performance of cEM improves with larger value of $\tau$ but also has the most variance compared to the other methods. For dEM and KTT, there exists an

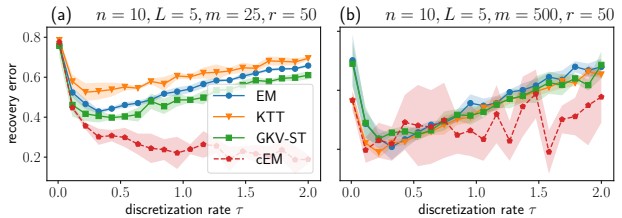

Figure 7: Recovery error for different discretization rates $\tau$: (a) 20 samples with 25-length trails and (b) 100 samples with 200-length trails.

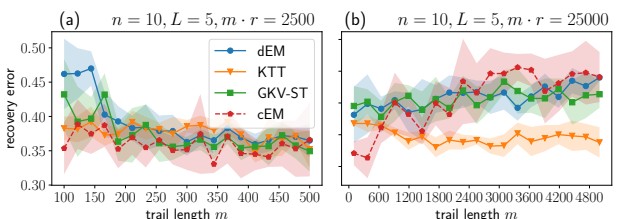

Figure 8: Recovery error based on transition counts: The plots present the error as a function of the total transitions, derived from multiplying the number of samples with the trail length. Two specific cases are highlighted: (a) 2 500 total transitions, and (b) 25 000 total transitions. The figure titles contain the remaining parameters, $\tau$ is set to 0.1.

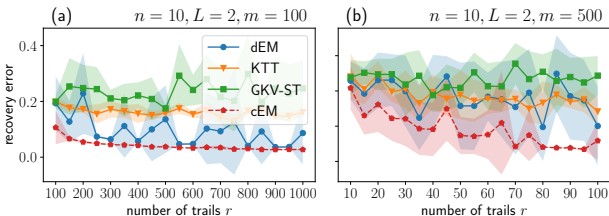

Figure 9: Recovery error under a varying number of trails. We consider trails of medium length in (a) and trails of long length in (b).

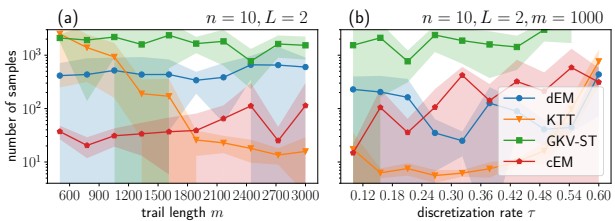

Figure 10: Sample complexity for a varying number of samples (a) and a varying discretization rate $\tau$ (b).

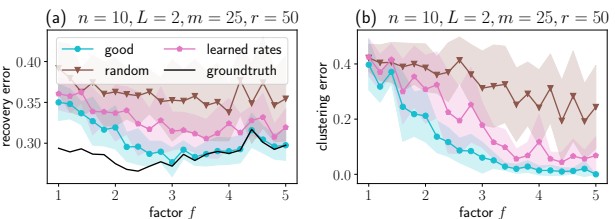

Figure 11: Mixture of $L = 2$ chains with $K^1 = f \cdot K^2$. We show the effect of different initializations, as explained in the text.

optimal value for $\tau$. We observe from Figures 7(a) and (b) that this $\tau$ value varies as we vary the trail length $m$. This optimal value strikes a balance between observing each transition sufficiently often to ensure effective clustering and keeping $\tau$ sufficiently small to achieve optimal recovery, represented by the mapping $e^{K\tau} \mapsto K$.

**Ranging the trail length $m$.** Figure 8 examines the behaviors of dEM, KTT, and GKV-ST, maintaining a consistent total number of observations $m \cdot r$ of (a) 2500 and (b) 25000. For longer trails, KTT surpasses even the cEM baseline. However, (a) also underscores dEM's reliance on adequately longer trails, a criterion not necessary for its continuous counterpart.

**Ranging the number of trails $r$.** Figures 9(a) and (b) show the recovery error for all methods as the number of samples increases from 100 to 1000 with a step of 50 and from 10 to 100 with a step of 5 respectively. The values for $n, L, m$ are 10, 2, and 500, respectively. We observe that cEM has the lowest recovery error in all cases. The performance of the other three methods alternates with dEM ranking as the second best. In the regime where we see few trails, the variance of all methods increases.

**Sample Complexity.** For dEM and KTT, we study the empirical sample complexity, i.e. the number of samples required to obtain a recovery error below a certain threshold. For our results, we use a threshold of 0.1. Figure 10 shows the sample complexity for varying trail length (a) and varying discretization rates $\tau$ (b). We can clearly observe that for increasing trail length, KTT performs better while dEM performs worse. This behavior is attributed to the local optimization nature of the EM algorithm. Furthermore, (b)

shows that KTT still achieves low recovery error if trails are long enough, even for low $\tau$, compared to dEM.

**Proportional Rates.** We consider the difficult case when the rate matrices $K^1$ and $K^2$ of a mixture are proportional, i.e. there exists a factor $f > 0$ such that $K^1 = f \cdot K^2$ for the same graph topologies. This is a difficult case as the discretization step may conflate the two chains into the same discrete chain. In this hard case, we found that dEM performed best. We thus use dEM with several initializations. For clarity, let us denote with $\mathcal{K}_{[0,f]}$ the uniform distribution over rate matrices $K$ with $K_{yz} \in [0, f]$ for states $y \neq z$. First, we initialize with a random mixture sampled from $(\mathcal{K}_{[0,1]}, \mathcal{K}_{[0,f]})$. We call this initialization good. Second, we try to learn the holding times first and initialize dEM with random rate matrices that have the learned holding times (learned). Third, we sample both initial rate matrices from $K_{[0,1]}$ (random). We observe that the recovery and clustering error is almost as good as when using the groundtruth clustering, after using the good initialization or learning the holding times.

**Scalability.** Figure 12 shows the running times of dEM, GKV-ST, KTT, and cEM. We vary the number of chains $L$ in (a) and the trail length $m$ in (b). KTT scales worse that the rest of the methods. The fastest is

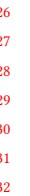
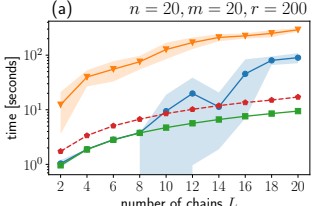
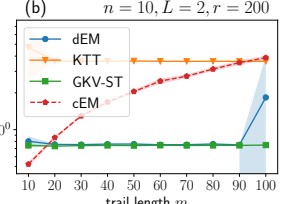

Figure 12: Running times for varying number of chains $L$ (b) and varying trail length (b).

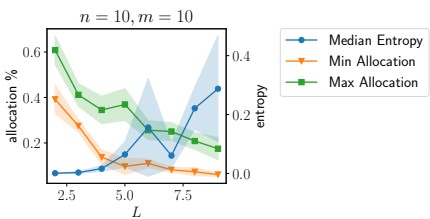

Figure 13: Allocation-percentage and median entropy for dEM on Last.fm.

based on the GKV-ST methodology for discrete chains [24, 40] with the caveat that the theoretical conditions of the recovery theorems may not always apply, as we observed in some of our experiments.

**Last.fm.** Figure 13 shows outcomes from supplementary experiments on the Last.fm dataset. We explore the learning of a mixture encompassing $L$ chains with values ranging from 2 to 10 by centering our attention on 20 users. This is done to discern typical user behaviors, also known as archetypical behaviors [16]. We graphically represent the median assignment entropy and the minimum and maximum of the assignment probabilities over the chains for dEM. In this context, the probability of assignment to a chain $\ell \in [L]$ is defined as $\frac{1}{r}\sum_{\mathbf{x}\in\mathbf{X}} a(\mathbf{x},\ell)$. We notice that as the number of chains increases, the entropy also tends to rise. This suggests that several chains produce comparable likelihoods for the trails, leading to greater uncertainty in assignment. As previously noted in the main content, determining a strategy for selecting $L$ remains an open challenge for future work.

**NBA.** We also provide additional qualitative results on the NBA dataset. Since hit and miss are the sole absorbing states, $\pi_K(\text{hit})$ and $\pi_K(\text{miss}) = 1 - \pi_K(\text{hit})$, they signify the odds of scoring or not scoring points, respectively. Thus, we can ascribe a score likelihood to each tactic. To gauge the efficacy of the deduced mixture, we assess its predictive precision, as done in other work [12]. With a given mixture of continuous-time Markov chains and a trail prefix $x' = (x_t)_{0 \le t \le t'}$ halting prior to reaching the absorbing states hit or miss at time $t'$, we ascertain the probabilities $\Pr[x' \cap \ell \mid \mathbf{K}]$ and $\Pr[x_\infty = \text{hit} \mid x' \cap K^\ell]$ for every $\ell \in [L]$, facilitating the determination of the score likelihood $\Pr[x_\infty = \text{hit}]$ via the theorem of total probability. We use trails with a 80%-20% train-test split from the teams Golden State Warriors (GSW), Boston Celtics (BOS), Los Angeles Lakers (LAL), Miami Heat (MIA), Los Angeles Clippers (LAC), and Houston Rockets (HOU) in the 2022 season. We plot the predictive accuracy of the chains learned via dEM, KTT, and cEM.

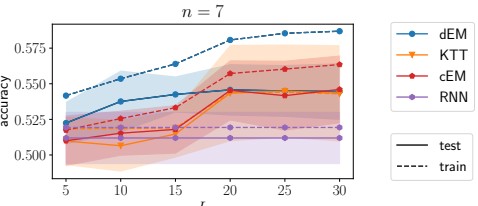

Figure 14: Train and test accuracy of miss and hit prediction using dEM, cEM, KTT and RNNs on the NBA dataset.

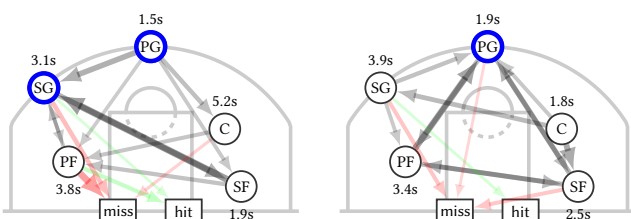

Figure 15: Two additional Golden State Warriors offensive tactics from the mixture in Figure 4. Based on our derived CTMCs, both tactics (left and right) have a scoring probability of 41%.

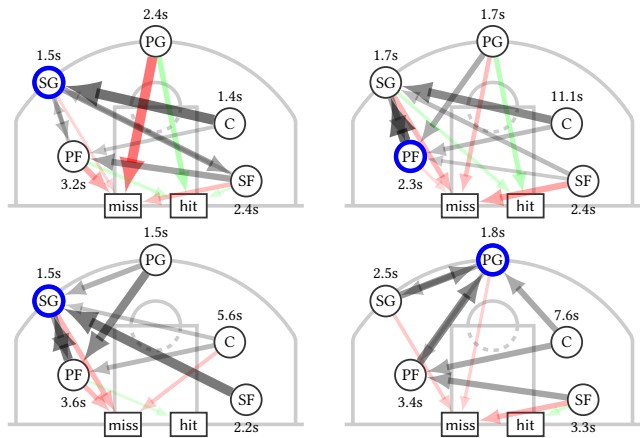

Figure 16: Offensive strategies for the New York Knicks represented as a mixture of $L = 4$ CTMCs. Parameter $\tau$ was set equal to 0.1 secs. The scoring probabilities are as follows: 37% (top left), 36% (top right), 35% (bottom left), and 40% (bottom right).

As a baseline, we implemented a recurrent neural network using Pytorch [23] that is trained on the set of discretized trails.

Finally, we show two additional offensive strategies for the Golden State Warriors (Figure 15) and another set of 4 offensive strategies learned from trails of the New York Knicks in the 2022 season in Figure 16. We will include more strategies from all teams in an extended journal version of our work.

Received 20 February 2007; revised 12 March 2009; accepted 5 June 2009

