# OpenReview forum: "Learning Mixtures of Continuous-Time Markov Chains"
_ACM.org/TheWebConf/2024/Conference — TheWebConf24_

### Official Review · Reviewer_h4vX · 2023-11-09

**Novelty:** 5
**Technical Quality:** 5

**Review:**

This paper introduces a novel framework for learning mixtures of continuous-time Markov chains, focusing on the influence of observed trails' length and mixture parameters on problem regimes which demand specific algorithms.

Pros\
S1: The paper tackles a significant and challenging problem in the stochastic processes domain, specifically the learning of mixtures of CTMCs, which has practical implications in various fields.\
S2: The authors conduct extensive experiments to validate their approach and provide a comparative analysis with state-of-the-art methods, which indicates a comprehensive empirical evaluation.\
S3: The paper makes theoretical contributions using advanced probabilistic tools and provides proofs, ensuring transparency and reproducibility of the results.

Cons\
W1: It is not clear how the proposed framework performs in terms of computational efficiency, especially when compared to existing methods.\
W2: The work is quite specialized, and its implications outside the domain of learning CTMCs are not discussed, which may limit the perceived applicability of the research.\
W3: There is no discussion on the limitations of the proposed framework or scenarios where it may not perform well, which is essential for readers to understand the scope of the methods.

**Questions:**

See W1-W3

**Reviewer Confidence:**

2: The reviewer is willing to defend the evaluation, but it is likely that the reviewer did not understand parts of the paper

**Scope:**

3: The work is somewhat relevant to the Web and to the track, and is of narrow interest to a sub-community

---

### Official Review · Reviewer_aKcy · 2023-11-09

**Novelty:** 5
**Technical Quality:** 6

**Review:**

This paper investigates learning discrete-time mixture for any specified discretization interval under certain lenient conditions in stochastic processes, which addresses the time-continuous (sequence) learning problem. This research introduced a novel framework for exploring CTMCs, emphasizing the influence of observed trails’ length and mixture parameters on problem regimes, which is further evaluated on the Synthetic, music as well as basketball dataset.

Strengths.
- S1. Good reproducibility.
- S2. Well theoretical guarantee.

**Questions:**

Q1. Please clarify the relationships between this work and the specific web applications, especially explain why the motivation is utilizing the datatset of NBA basketball.

Q2. This paper mainly investigates the Mixtures of Continuous-Time Markov Chains, which is faced with a limited range of audience. What other applications, especially for the web applications should be discussed.

**Ethics Review Description:**

N/A.

**Reviewer Confidence:**

2: The reviewer is willing to defend the evaluation, but it is likely that the reviewer did not understand parts of the paper

**Scope:**

2: The connection to the Web is incidental, e.g., use of Web data or API

---

### Official Review · Reviewer_DaZM · 2023-11-12

**Novelty:** 6
**Technical Quality:** 6

**Review:**

This paper focuses on continuous-time Markov chains (CTMCs) and introduces a novel framework framework that emphasizes on the influence of observed trails’ length. The proposed algotithm comprises three stages: discretization, soft clustering, and recovery to learn mixtures of CTMCs. Analytical and experimental results offer valuable insights to process continuous-time problems.

The paper is generally well written and provides comprehensive theoratical analysis. And the proposed algorithm has shown effectiveness based on comprehensive experiments from synthetic and real-world datasets. However, one thing this reviewer is concerned is the method is tested on relatively limited scale datasets. It would be more persuasive to test on larger datasets such as Last.fm-360k.

**Questions:**

Please refer to the review.

**Reviewer Confidence:**

2: The reviewer is willing to defend the evaluation, but it is likely that the reviewer did not understand parts of the paper

**Scope:**

3: The work is somewhat relevant to the Web and to the track, and is of narrow interest to a sub-community

---

### Official Review · Reviewer_19rp · 2023-11-19

**Novelty:** 3
**Technical Quality:** 4

**Review:**

In this paper, the authors delve into the theoretical aspects of handling "learning mixtures of homogeneous Continuous-Time Markov Chains (CTMCs) from trail data." They offer formal definitions regarding background theories, the proposed algorithm, and methods. The effectiveness of their approach was evaluated through experiments conducted on popular datasets, employing various customized algorithms.

Primarily theoretical, the authors highlight the usage of Continuous-Time Markov Chains (CTMCs) in modeling biological, financial behaviors, and evolution. However, the connection between the proposed approach and web-related problems remains unclear, as emphasized in the paper.

The authors assert their research goal of addressing "learning mixtures of homogeneous CTMCs from trail data." To elucidate this goal, clearer background information and definitions are necessary. Understanding the concept of "learning mixtures," homogeneity factors of CTMCs, and defining "trail data" becomes crucial. Instead of solely presenting the formal representation of Markov chains, an elucidation of the research's motivation, practical problems to solve, and real-world applications, especially related to the web, would enhance comprehension.

The related work section primarily offers background information on Markov chains but lacks discussions on state-of-the-art techniques addressing similar problems.

Within the experiment section, it's imperative to expound on the rationale behind selecting the Last.fm and NBA datasets and demonstrate their representativeness in evaluating the proposed approach. Additionally, clarification is needed on why the implementation of the three customized algorithms—GKV-ST, dEM, and KTT—was crucial in showcasing real-world efficiency. Moreover, the reasoning behind utilizing cEM for comparative analysis needs clarification.

The significance of the three research questions introduced in Section 4 requires more elaboration. Specifically, explaining the concept of "practical boundaries of the problem regimes" in response to the first question is essential. Linking the research questions explicitly to the experiment results and discussing new insights derived from the analysis results are imperative for clarity and comprehensiveness.

**Questions:**

Why were the three customized algorithms, GKV-ST, dEM, and KTT, chosen to showcase real-world efficiency?

What prompted the use of cEM for comparative analysis?

What is the concept of "practical boundaries of the problem regimes" and what is its significance in evaluating the approach?

What novel insights emerge from the analysis results?

**Reviewer Confidence:**

2: The reviewer is willing to defend the evaluation, but it is likely that the reviewer did not understand parts of the paper

**Scope:**

2: The connection to the Web is incidental, e.g., use of Web data or API

---

### Official Review · Reviewer_t5MK · 2023-11-24

**Novelty:** 4
**Technical Quality:** 5

**Review:**

The paper addresses the problem of learning homogeneous CMTCs (Continuous-Time Markov Chains) from trail data, demonstrating the  combined impact of trails' length and the mixture parameters on the problem nature. Examining the impact of discretization of continuous-time trails on the learnability of continuous-time mixture, the authors extend the MLE (Maximum Likelihood Estimation), and propose a framework with: (1) three
basic stages: discretization, soft clustering, and recovery; (2) - a recommendation of appropriate algorithm to be used (three practical implementations - GKV-ST, dEM and KTT - are presented for respective theoretical guarantees). Experimental evaluations are presented quantitatively addressing 3 questions regarding: how trail length affects soft clustering impact on the algorithmic performance; the accuracy of soft clustering and its impact on recovery error; impact of recovery on th error. Three kinds of datasets are considered (synthetic; Last.fm (user music history); NBA data regarding ball passing) evaluating the behavior of GKV-ST, dEM and KTT w.rt. recovery error, runtime and clustering error; classification error and assignment entropy. A lot more experiments are presented in the Appendix, along with theoretical derivations.

S1: Interesting idea of developing a learning-based framework to recommend appropriate algorithm.

S2: Rather clear presentation and well-organized theoretical results (justified in the Appendix).

S3: Using of novel dataset from NBA passing is an intriguing application of CMTC.


W1: The authors should have made it more clear as to what is the "punchline" in terms of the technical novelty. It is hard to assess where one draws the line between relying on combining existing methods for a practical tool vs. novel methods.

W2: It seems like the only baseline used in the main body of the paper is cEM. The authors should have used more detailed comparison with more baselines.

w3: The biggest problem that this reviewer has stems from the observation that the scope of the work is not appropriate for TheWebConference. Namely, the call for papers is very clear:
"A typical Web Conference paper should have explicit focus on at least one of the following:

- understanding, evaluating, and improving the Web as a technical infrastructure; including core Web technologies, standards, and platforms
- understanding, evaluating, and improving the Web as a socio-economic system;
- understanding better the impact of the Web and Web technologies;
- democratizing access to Web content and technologies, making it more accessible, fair, inclusive, and accountable to a wide range of audiences."

and there are a few Tracks. However, aside from mentioning "web" exactly once in the abstract, the main "story" is motivated by the importance of CTMC in: "biological systems...; finance...; molecular kinetics...". Similarly, the experimental evaluation is strictly focused on distinct application domains but are not quite "weaved" into the desiderata for the web conference and it is not to easy to assess the association of the work with a particular Track.

NOTE: This is not to say that the paper could not be a good contribution to other forum (e.g., ADS Track at KDD)

**Questions:**

Please refer to W1-W3 above.

**Ethics Review Description:**

No ethics-related problems.

**Reviewer Confidence:**

3: The reviewer is confident but not certain that the evaluation is correct

**Scope:**

1: The work is irrelevant to the Web

---

### Official Review · Reviewer_Chg9 · 2023-12-01

**Novelty:** 5
**Technical Quality:** 5

**Review:**

The paper focuses on the problem of learning mixtures of continuous-time Markov chains (CTMCs).
The authors introduce a new framework for learning mixtures of CTMCs, through both continuous and discrete-time observations and methods to tailor it depending on the length of the trails.
Extensive experiments on two real-world datasets (Lastfm and NBA) demonstrate the effectiveness of the proposed solution.




(+) The motivation of the paper is reasonable.

(+) The authors conduct experiments on real datasets to demonstrate the effectiveness of the proposed solution.

(+) The code is available.


(-) The survey is somewhat limited.

(-) The technical depth of the work is medium.

(-) The technical contribution of the paper is relatively weak.



The paper focuses on an important problem.
The authors have conducted various comparisons and experiments to prove their claim.
They have compared their method with state-of-the-art methods, including one very recent work (WWW'23[40], PMLR'23[28]), which makes the paper more reliable.

The idea of using CTMCs to capture the dynamics of continuous-time trails looks interesting.
It seems to be more explanatory than traditional Markov chain models and their variations (e.g., [24][40]).

The survey is somewhat limited.
The authors provide a detailed survey of the related work, mainly focusing on Markov chain models.
I would like to see more discussion about other kinds of dynamic/time-series modeling approaches.

The technical contribution of the paper seems to be relatively weak since most of the proposed method's design is similar to the existing approaches (e.g.,  CTMCs, [24][40]).
I think it would be better to emphasize the novelty of the method more clearly.

The definition of (continuous-time/observed) trails is somewhat unclear:
I think providing brief explanations (or some running examples) in the introduction would be helpful.

**Questions:**

N/A

**Reviewer Confidence:**

2: The reviewer is willing to defend the evaluation, but it is likely that the reviewer did not understand parts of the paper

**Scope:**

3: The work is somewhat relevant to the Web and to the track, and is of narrow interest to a sub-community

---

### Decision · Program_Chairs · 2024-01-22

**Decision:**

Accept

**Comment:**

Summary: A framework for studying mixtures of continuous-time Markov chains.


 Strengths:
 + Addresses an interesting/real-world problem
 + Novel framework and algorithms
 + Some theoretical analyses and proofs
 + Comprehensive experiments


 Weaknesses:
 - Related work could be improved
 - Technical novelty and depth moderate
 - More and better baselines needed
 - Could experiment on larger-scale datasets
 - Computational efficiency
 - Some concerns on relevance to the conference


 Recommendation: Good theoretical contribution